# Serine 25 phosphorylation inhibits RIPK1 kinase-dependent cell death in models of infection and inflammation

Yves Dondelinger[1,2], Tom Delanghe[1,2], Dario Priem[1,2], Meghan A. Wynosky-Dolfi [3], Daniel Sorobetea[3], Diego Rojas-Rivera[1,2,8], Piero Giansanti[4,5,9], Ria Roelandt[1,2], Julia Gropengiesser[6], Klaus Ruckdeschel [6], Savvas N. Savvides [1,7], Albert J.R. Heck [4,5], Peter Vandenabeele [1,2], Igor E. Brodsky [3] & Mathieu J.M. Bertrand [1,2]

RIPK1 regulates cell death and inflammation through kinase-dependent and -independent mechanisms. As a scaffold, RIPK1 inhibits caspase-8-dependent apoptosis and RIPK3/MLKL-dependent necroptosis. As a kinase, RIPK1 paradoxically induces these cell death modalities. The molecular switch between RIPK1 pro-survival and pro-death functions remains poorly understood. We identify phosphorylation of RIPK1 on Ser25 by IKKs as a key mechanism directly inhibiting RIPK1 kinase activity and preventing TNF-mediated RIPK1-dependent cell death. Mimicking Ser25 phosphorylation (S > D mutation) protects cells and mice from the cytotoxic effect of TNF in conditions of IKK inhibition. In line with their roles in IKK activation, TNF-induced Ser25 phosphorylation of RIPK1 is defective in TAK1- or SHARPIN-deficient cells and restoring phosphorylation protects these cells from TNF-induced death. Importantly, mimicking Ser25 phosphorylation compromises the in vivo cell death-dependent immune control of *Yersinia* infection, a physiological model of TAK1/IKK inhibition, and rescues the cell death-induced multi-organ inflammatory phenotype of the SHARPIN-deficient mice.

[1] VIB Center for Inflammation Research, 9052 Ghent, Belgium. [2] Department of Biomedical Molecular Biology, Ghent University, 9052 Ghent, Belgium. [3] Department of Pathobiology, School of Veterinary Medicine, University of Pennsylvania, Philadelphia, PA 19104, USA. [4] Biomolecular Mass Spectrometry and Proteomics, Bijvoet Centre for Biomolecular Research and Utrecht Institute for Pharmaceutical Sciences, University of Utrecht, 3584 CH Utrecht, The Netherlands. [5] Netherlands Proteomics Centre, 3584 CH Utrecht, The Netherlands. [6] Institute for Medical Microbiology, Virology and Hygiene, University Medical Center Eppendorf, 20246 Hamburg, Germany. [7] Department of Biochemistry and Microbiology, Ghent University, 9052 Ghent, Belgium. [8] Present address: Center for Integrative Biology (CIB), Faculty of Sciences, Universidad Mayor, 8580745 Santiago, Chile. [9] Present address: Chair of Proteomics and Bioanalytics, Technical University of Munich, 85354 Freising, Germany. These authors contributed equally: Yves Dondelinger, Tom Delanghe. Correspondence and requests for materials should be addressed to M.J.M.B. (email: mathieu.bertrand@irc.vib-ugent.be)

Receptor Interacting Protein Kinase 1 (RIPK1) has emerged as a major signaling hub downstream of several immune receptors, where it regulates cell death and inflammation through kinase-dependent and -independent mechanisms[1]. As a scaffold molecule, RIPK1 facilitates activation of the MAPK and NF-κB pathways and inhibits caspase-8-dependent apoptosis and RIPK3/MLKL-dependent necroptosis. On the other hand, as a kinase, RIPK1 paradoxically induces apoptosis and necroptosis following its enzymatic activation. The fact that RIPK1-deficient mice die perinatally, while mice endogenously expressing a catalytically inactive version of RIPK1 reach adulthood without developing any spontaneous overt phenotype, demonstrates the predominant pro-survival scaffolding role of RIPK1 during development[2–4]. Nevertheless, RIPK1 kinase-dependent cell death has revealed its importance in the context of host-pathogen interactions, where it can either participate in the control of infection or favor it[5–8]. In addition, RIPK1 kinase-dependent cell death has also been demonstrated to drive the pathogenesis of various inflammatory diseases in mice, which motivated the recent clinical trials for the potential therapeutic use of RIPK1 kinase inhibitors in human[9–11]. Despite these exciting advances, the precise molecular mechanism regulating the switch between RIPK1 pro-survival and pro-death functions has remained poorly understood.

RIPK1 is most extensively studied in the context of TNF signaling. Binding of TNF to TNFR1 results in the rapid assembly of a receptor-bound primary complex (complex I) that includes, among others, RIPK1, TRADD, cIAP1/2, LUBAC (composed of SHARPIN, HOIP and HOIL-1), TAB-TAK1, and the IKK complex (composed of NEMO, IKKα, and IKKβ). A network of polyubiquitin chains generated by cIAP1/2 and LUBAC tightly controls the stability of complex I and the ability of the receptor to activate the MAPK and NF-κB signalling pathways[12,13]. These ubiquitin chains, conjugated to RIPK1 and other components of complex I, generate binding sites for the adaptor proteins TAB2/3 and NEMO, which, respectively, recruit TAK1 and IKKα/β to the complex, and ultimately lead to gene expression via downstream activation of the MAPK and NF-κB pathways[14,15]. RIPK1 kinase-dependent cell death is not the default response of most cells to TNF sensing. It generally requires further inactivation of transcription-independent molecular checkpoints that prevent RIPK1 from promoting, in a kinase-dependent way, the assembly of a secondary cytosolic complex that either triggers caspase-8-mediated apoptosis (complex IIb) or RIPK3/MLKL-mediated necroptosis (necrosome)[16,17]. The ubiquitin chains conjugated to RIPK1 by cIAP1/2 and LUBAC in complex I have been reported to repress RIPK1 cytotoxic potential, both directly as well as indirectly by promoting p38/MK2-, TBK1/IKKε-, and IKKα/β−phosphorylation of RIPK1[18–26]. While TBK1/IKKε- and IKKα/β-phosphorylation of RIPK1 represents a critical brake in the TNFR1 death pathway, phosphorylation by MK2 only serves as a second layer of protection that limits the extent of cell death in killing conditions[27]. The role of IKKα/β in repressing RIPK1 cytotoxicity is NF-κB-independent, and its physiological importance is demonstrated by the fact that inflammatory pathologies caused by IKKα/β inactivation in mice can be driven by RIPK1 kinase-dependent cell death[22,28]. Defects in this IKKα/β checkpoint presumably also explain, at least in part, the in vivo inflammatory phenotypes caused by RIPK1 kinase-dependent cell death in conditions affecting proper expression/activity of IKKα/β upstream activators, such as in NEMO-deficient mice[29,30], SHARPIN-deficient mice[3], or mice in which TAK1/IKKs are inhibited following *Yersinia* infection[6]. How exactly IKKα/β-phosphorylation of RIPK1 prevents RIPK1 kinase-dependent death has, however, so far remained unanswered.

In this study, we identify IKKα/β−mediated phosphorylation of RIPK1 on Ser25 as a physiological brake that directly inhibits RIPK1 kinase activity and prevents TNF-mediated RIPK1 kinase-dependent cell death. We therefore report on a precise molecular mechanism controlling the switch between RIPK1 pro-survival and pro-death functions and demonstrate its physiological relevance in mouse models of infection and inflammation.

## Results

**IKKα/β phosphorylate RIPK1 on Ser25 in TNFR1 complex I**. We previously reported that RIPK1 is a direct substrate of both IKKα and IKKβ, and that the simultaneous inactivation of IKKα and IKKβ affects RIPK1 phosphorylation in TNFR1 complex I and switches the TNFR1 response from survival to RIPK1 kinase-dependent cell death[22]. To better understand this NF-κB-independent protective role of IKKα/β, we performed mass spectrometry (LC-MS/MS) analysis of complex I isolated from WT and *Ikkα/β*[−/−] mouse embryonic fibroblasts (MEFs) in the hope to identify residues of RIPK1 phosphorylated by these kinases in response to TNF (Fig. 1a, b). The LC-MS/MS analysis generated a list of proteins containing phosphorylated residues (Supplementary Data 1). Among them, we identified Ser6 and Ser25 of RIPK1 to be greatly hypo-phosphorylated in absence of IKKα/β, while Ser166 and Thr169, two autophosphorylation residues of RIPK1, were hyper-phosphorylated (Fig. 1c). Phosphorylation on Ser330 was, on the other hand, unaffected by the absence of IKKα/β. Together with the fact that Ser25 was previously identified by LC-MS/MS as a direct substrate of IKKs in a kinase assay making use of recombinant IKKα/β and RIPK1[22], these results support direct phosphorylation of RIPK1 on Ser25, and potentially on Ser6, by IKKα/β in TNFR1 complex I. To evaluate whether the reported lethal switch in the TNF response obtained in IKKα/β inactivated conditions could originate from defective phosphorylation of RIPK1 on any of these two residues, we reconstituted RIPK1-deficient MEFs with wild-type (WT) or the phospho-mimetic S > D mutated versions of RIPK1, and stimulated the cells with TNF in presence of IKK inhibitor (IKKi) and caspase inhibitor (zVAD) (Fig. 1d). Remarkably, restoring Ser25 phosphorylation, but not Ser6 phosphorylation, greatly protected the cells from RIPK1 kinase-dependent death, as demonstrated by the reconstitution of the MEFs with the K45A kinase-dead RIPK1 mutant or by the use of the RIPK1 kinase inhibitor Necrostatin-1s (Nec-1s) (Fig. 1d). Serine 25 of RIPK1 is highly conserved within mammals (Supplementary Fig. 1A), and we found that the phospho-mimetic S25D mutation similarly protected human Jurkat cells from the death trigger (Fig. 1e), indicative of a conserved regulatory mechanism.

Occurrence of IKKα/β-mediated Ser25 phosphorylation was further demonstrated by specific pS25 RIPK1 immunoprecipitations (IP) (Supplementary Fig. 1B) in WT and IKKα/β-inactivated MEFs, obtained either by pharmacological or genetic means (Fig. 1f–g). When visualizing RIPK1 by immunoblot after pS25 RIPK1 pull-down in WT MEFs, the signal appeared as a phosphorylation smear, which was collapsed following post-IP λ phosphatase treatment (Fig. 1f). These results therefore indicate multiple phosphorylation events on the same RIPK1 molecules, including on Ser25. Since MK2-mediated phosphorylation was excluded by performing the experiment in MK2-inhibited conditions, it could indicate that IKKα/β simultaneously phosphorylate RIPK1 on multiple sites. Interestingly, kinetics of pS25 RIPK1 IP showed that phosphorylation of RIPK1 on Ser25 was a dynamic process that peaked at 10 min of TNF stimulation (Fig. 1h). Finally, we confirmed occurrence of Ser25 phosphorylation of the pool of RIPK1 associated with TNFR1 complex I by

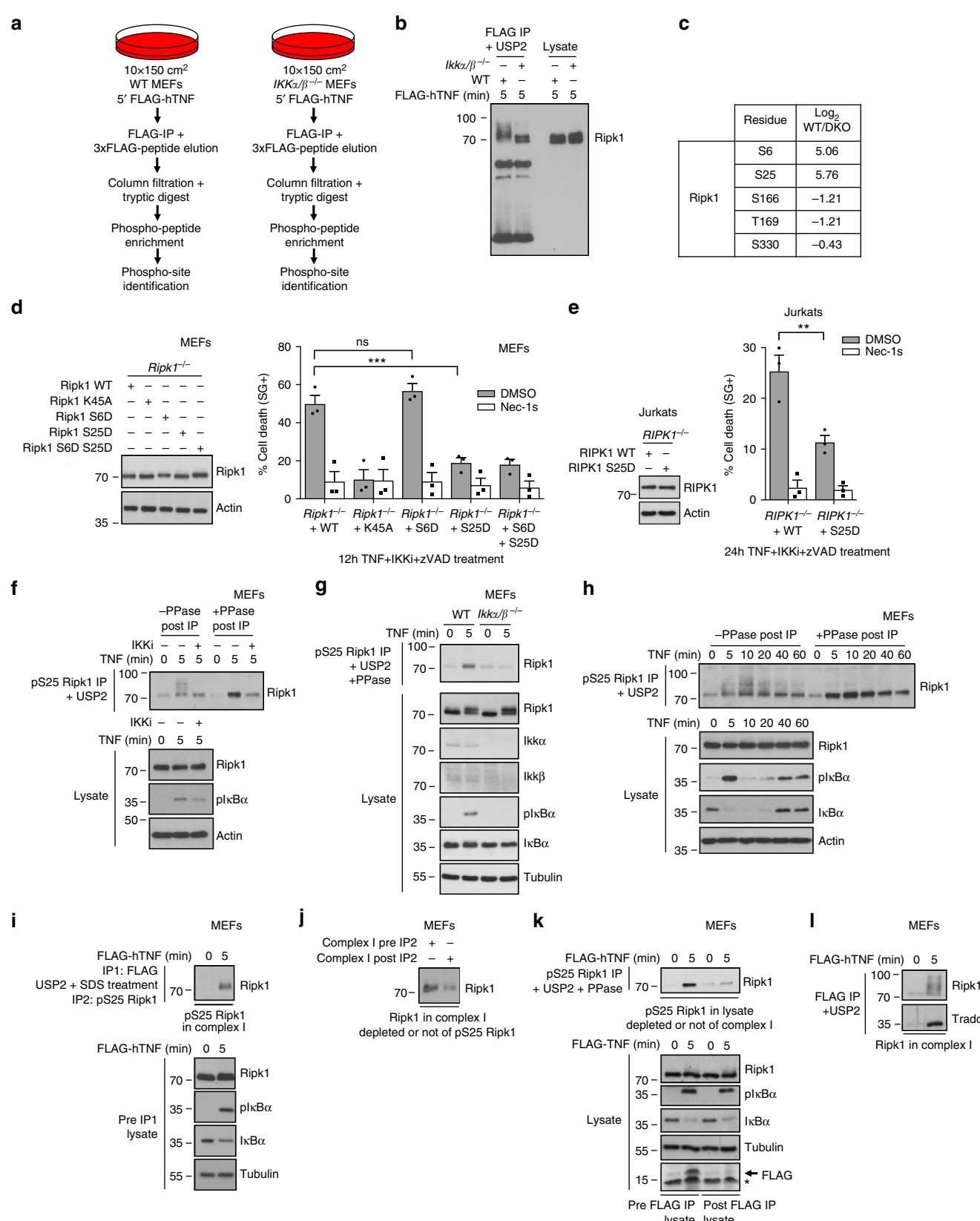

successively immunoprecipitating complex I and then pS25 RIPK1 from complex I (Fig. 1i). The analysis of the abundance of RIPK1 in complex I prior and after pS25 RIPK1 depletion also indicated that most of the RIPK1 associated with complex I is phosphorylated on Ser25 (Fig. 1j). In contrast, Ser25

phosphorylation was poorly detected in the pool of RIPK1 non-associated to complex I, which is visualized by analyzing the abundance of pS25 RIPK1 in the lysate prior and after complex I depletion (Fig. 1k–l). This indicates that phosphorylation of RIPK1 on Ser25 predominantly occurs in complex I.

**Fig. 1** IKKα/β phosphorylate RIPK1 on Ser25 in TNFR1 complex I. **a** Schematic overview of the mass spectrometry (LC-MS/MS) experiment. **b** Small-scale control for the LC-MS/MS experiment. Wild-type (WT) or *Ikkα/β−/−* MEFs were stimulated with 1 μg/ml FLAG-hTNF, TNFR1 complex I was FLAG-immunoprecipitated and treated with USP2 to reveal difference in RIPK1 phosphorylation profiles. **c** Relative abundance between WT and *Ikkα/β−/−* (DKO) MEFs of the phosphorylated residues of RIPK1 identified by LC-MS/MS. **d** *Ripk1−/−* MEFs and **e** *RIPK1−/−* Jurkat cells were reconstituted with the indicated RIPK1 constructs, pretreated or not with Nec-1s and ZVAD-fmk for 30 min and then stimulated with 1 ng/ml (**d**) or 20 ng/ml (**e**) of hTNF. Cell death was measured in function of time by SytoxGreen positivity and is presented as mean ± SEM of independent experiments (*n* = 3). Statistical significance was determined using two-way ANOVA followed by a Sidak (**d**) or Dunnett (**e**) post-hoc test. The significance between samples is indicated in the figures as follows: ∗∗*p* < 0.01; ∗∗∗*p* < 0.001; nsnon-significant. Wild-type (**f–l**) or *Ikkα/β−/−* (**g**) MEFs were pretreated with the indicated compounds for 30 min before stimulation with 1 μg/ml hTNF (**f–h**) or 1 μg/ml FLAG-hTNF (**i–l**) for the indicated duration. **f–h** pS25 RIPK1 was immunoprecipitated and treated with USP2 and λ phosphatase (PPase) post-IP when indicated. **i–j** TNFR1 complex I was purified by FLAG immunoprecipitation (IP1), the complex was then released from the beads by SDS treatment, and pS25 RIPK1 was subsequently immunoprecipitated (IP2) from the released complex I eluate. **k** pS25 RIPK1 was immunoprecipitated from lysates previously depleted or not of complex I by FLAG immunoprecipitation (**l**), and next treated with USP2 and PPase. (**b**, **d–l**) Protein expression levels were determined by immunoblot. Immunoblots are representative of 1 (**d**, **e**, **h**) or 2 (**b**, **f**, **g**, **i–l**) independent experiments

**Mimicking phospho-Ser25 protects from TNF-induced cell death**. To further address the physiological relevance of IKKα/β-mediated pS25 RIPK1, we generated a phospho-mimetic S25D RIPK1 knockin mouse line using the CRISPR/Cas9 technology (Supplementary Fig 2A–B). The *Ripk1S25D/S25D* mice were viable, healthy and did not develop any spontaneous overt phenotype (Supplementary Fig. 2C–D). RIPK1 expression levels were also indistinguishable in the organs of *Ripk1+/+* and *Ripk1S25D/S25D* littermates (Supplementary Fig. 2E), indicating that the S25D mutation has no impact on RIPK1 stability. TNF stimulation of bone marrow-derived macrophages (BMDMs) or MEFs isolated from these mice revealed no difference in the ability of the mutated RIPK1 to activate the MAPK and NF-κB pathways (Supplementary Fig. 3A–B). Instead, restoring pS25 RIPK1 with the S25D mutation greatly protected BMDMs and MEFs from TNF-induced RIPK1 kinase-dependent apoptosis and necroptosis induced by IKK inhibition, respectively, in the absence or presence of zVAD (Fig. 2a–g). Of note, single allele mutation of RIPK1 provided partial protection to the death trigger (Supplementary Fig. 3C, D). The protection was associated with reduced processing of caspases (Fig. 2c), phosphorylation of MLKL (Fig. 2f), assembly of complex IIb/necrosome (Fig. 2g), and autophosphorylation of RIPK1 on Ser166—used as a readout for RIPK1 kinase activation (Fig. 2c–g). Importantly, the phospho-mimetic S25D mutation did not protect the cells from TNF-induced RIPK1-independent apoptosis obtained by co-stimulation with the translational inhibitor cycloheximide (CHX)[31,32], thereby demonstrating the specificity of the protection to conditions inducing RIPK1 kinase-dependent cell death (Supplementary Fig. 3E–G). We previously reported that in vivo administration of IKKi together with sublethal doses of TNF results in drastic hypothermia and lethality caused by RIPK1 kinase-dependent apoptosis and necroptosis[22]. Remarkably, both *Ripk1S25D/+* and *Ripk1S25D/S25D* mice showed protection to this lethal challenge (Fig. 2h–j). Of note, these mice were also greatly protected from a lethal dose of TNF injected in absence of IKKi (Fig. 2k–l), a model of Systemic Inflammatory Response Syndrome (SIRS) also caused by RIPK1 kinase-dependent apoptosis and necroptosis[3,9,33]. Together, these results highlight the crucial protective role of IKK-mediated pS25 RIPK1 against TNF-induced RIPK1 kinase-dependent cell death in vitro and in vivo.

**Phospho-Ser25 regulates the immune response against *Yersinia***. TAK1 inactivation is also known to switch the TNF response from survival to RIPK1 kinase-dependent death[22,32]. Because TAK1 is an upstream activator of IKKα/β, we then evaluated whether the cell death induced upon TAK1 inhibition also originates from defective pS25 RIPK1. In accordance with this idea, we found that pharmacological inhibition of TAK1 (TAK1i)

affected TNF-induced pS25 RIPK1 (Fig. 3a), and that restoring pS25 RIPK1 with the S25D mutation protected BMDMs and MEFs from TNF-induced RIPK1 kinase-dependent apoptosis when TAK1 is inhibited (Fig. 3b–d). Interestingly, we found that mimicking pS25 RIPK1 also protected BMDMs from death induced by LPS in presence of IKK or TAK1 inhibition (Fig. 3e–f). The mammalian pathogenic species of the Gram-negative *Yersinia* genus inject an acetyltransferase, named YopJ/P, into target cells. YopJ/P inhibits the catalytic activity of IKKs and TAK1 in an attempt to escape host defenses by preventing NF-κB- and MAPK-dependent expression of pro-inflammatory mediators[34–38]. In response to this hijacking, TLR4/TNFR1-mediated RIPK1 kinase-dependent apoptosis of the host cell has evolved as a backup mechanism providing cell-extrinsic signals to promote optimal antibacterial immunity[6,21,39,40]. In line with the defect of TNF-induced pS25 RIPK1 observed in IKKs or TAK1 inhibited conditions (Figs. 1f–g, 3a), we found that mimicking pS25 RIPK1 protected BMDMs from YopJ/P-dependent apoptosis following *Y. enterocolitica* (expressing YopP) and *Y. pseudotuberculosis* (expressing YopJ) infection (Fig. 3g–h). Of note, the protection observed in the *Ripk1S25D/S25D* BMDMs was comparable to the one obtained in cells where RIPK1 kinase activity was inhibited pharmacologically (Nec-1s) or genetically (*Ripk1K45A/K45A*) (Fig. 3g–i). Since RIPK1 kinase-dependent apoptosis of hematopoietic cells was shown to be required for optimal in vivo control of *Y. pseudotuberculosis* infection[6], bone marrow (BM) chimeras were generated by reconstituting lethally irradiated congenic hosts with BM from either *Ripk1+/+*, *Ripk1S25D/S25D*, or *Ripk1K45A/K45A* mice (Fig. 3j). Remarkably, *Ripk1S25D/S25D* and *Ripk1K45A/K45A* chimeras had increased bacterial burden in the liver and spleen when compared with WT BM chimeras, which resulted in reduced viability (Fig. 3k–m). Together, these results indicate that inhibition of hematopoietic Ser25 phosphorylation of RIPK1 by YopP/J activates a backup mechanism consisting in RIPK1 kinase-dependent cell death that is required for proper immunity to *Yersinia* infection.

**Defective phospho-Ser25 can drive multi-organ inflammation**. Chronic proliferative dermatitis in mice (cpdm) is a multi-organ inflammatory disorder caused by lack of SHARPIN expression and resulting from tissue specific induction of TNF/TNFR1-mediated RIPK1 kinase-dependent apoptosis or necroptosis[3,41–44]. In light of the reported role of SHARPIN in TNFR1-mediated IKKα/β activation[44], we found that TNF-induced pS25 RIPK1 was defective in mouse dermal fibroblasts (MDFs) isolated from SHARPIN-deficient mice (*Shpncpdm/cpdm*) (Fig. 4a), and that restoring Ser25 phosphorylation with the S25D mutation completely protected these cells from TNF-induced RIPK1 kinase-dependent apoptosis (Fig. 4b, c). More importantly, crossing the *Ripk1S25D/S25D* mice with the *Shpncpdm/cpdm*

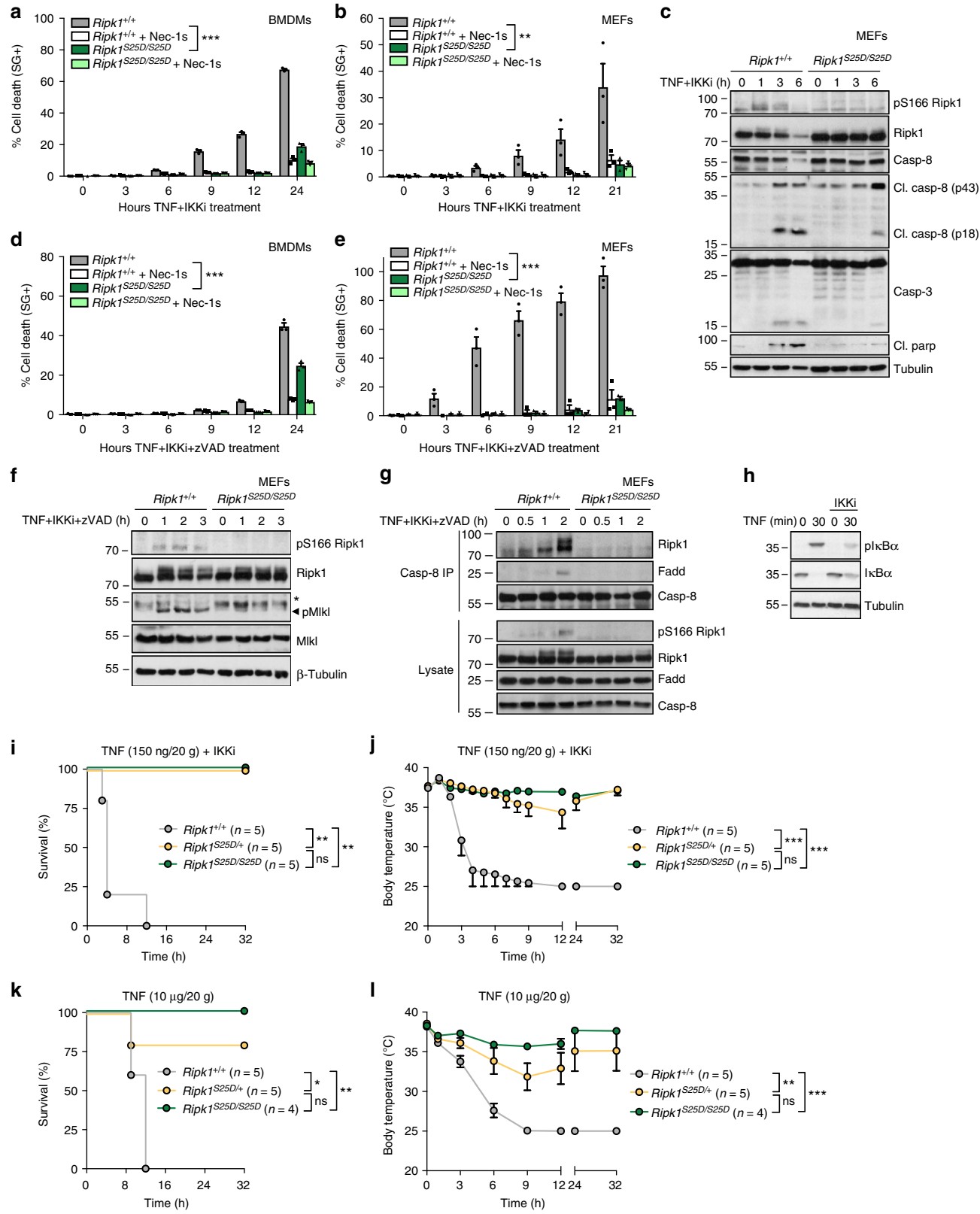

mice completely rescued the cell death-driven multi-organ inflammatory phenotype of the $Shpn^{cpdm/cpdm}$ mice (Fig. 4d–h). For example, the phospho-mimetic S25D mutation prevented the reported apoptosis-driven skin lesions and necroptosis-driven splenomegaly caused by lack of SHARPIN expression, as demonstrated by macroscopic analysis and staining of tissue sections with H&E, TUNEL (cell death) and cleaved Caspase-3 (apoptosis) (Fig. 4d), and by the weight of the spleens (Fig. 4e). The rescue of the phenotype is also demonstrated by the normalization in the serum levels of the cell death marker LDH (Fig. 4e), of the pro-inflammatory cytokine IL-6 (Fig. 4g), and of the chemokine MCP-1 (Fig. 4h).

**Fig. 2** Mimicking phospho-Ser25 protects from TNF-induced cell death. **a–g** *Ripk1*[+/+] and *Ripk1*[S25D/S25D] BMDMs (**a**, **d**) or MEFs (**b**, **c**, **e–g**) were pretreated with the indicated compounds for 30 min before stimulation with hTNF (1 ng/ml for BMDMs (**a**, **d**) and 100 pg/ml (**b**, **e**) or 20 ng/ml (**c**, **f**, **g**) for MEFs). Activation of cytosolic proteins was monitored by immunoblotting (**c**, **f**, **g**) and cell death was measured in function of time by SytoxGreen positivity (**a**, **b**, **d**, **e**). Cell death data are presented as mean ± SEM of three independent experiments ($n = 3$). BMDMs results were obtained with cells isolated from three different mice for each genotype ($n = 3$) (**a**, **d**). Statistical significance for the cell death assays was determined using two-way ANOVA followed by a Tukey post-hoc test. **h** In order to demonstrate efficacy of the IKKi when administered in vivo, mice were i.v. injected with IKKi (200 µg/20 g) 15 min prior to i.p. injection of mTNF (1 µg/20 g) in WT mice. Liver lysates were prepared and protein levels were determined by immunoblot. **i–j** *Ripk1*[+/+], *Ripk1*[S25D/+], and *Ripk1*[S25D/S25D] mice were i.v. injected with IKKi (200 µg/20 g) 15 min prior to i.v. injection of mTNF (150 ng/20 g). **k–l** *Ripk1*[+/+], *Ripk1*[S25D/+], and *Ripk1*[S25D/S25D] mice were i.v. injected with mTNF (10 µg/20 g). Cumulative survival rates (**i**, **k**) and body temperature (**j**, **l**) were determined in function of time. The number of mice (**n**) used in each condition is indicated. The temperature results are represented as mean ± SEM. Statistical significance for body temperatures of the mice were determined using two-way ANOVA followed by a Tukey post-hoc test. Survival curves were compared using log-rank Mantel–Cox test. Significance between samples is indicated in the figures as follows: ∗$p < 0.05$; ∗∗$p < 0.01$; ∗∗∗$p < 0.001$; ns non-significant. Immunoblots are representative of 2 (**c**, **f**, **g–h**) independent experiments

**Phospho-Ser25 prevents RIPK1 kinase activation in complex I.**
The current model suggests that TNF-mediated RIPK1 kinase-dependent cell death requires RIPK1 recruitment to complex I, followed by its dissociation from TNFR1 and finally its autophosphorylation-dependent association with FADD to initiate formation of the cytosolic complex IIb/necrosome[31,32,45–48]. Although low levels of RIPK1 enzymatic activity, monitored by autophosphorylation on Ser166, have been reported in TNFR1 complex I[49], it is currently unclear whether the boost in RIPK1 kinase activity associated with RIPK1 kinase-dependent cell death occurs in complex I or only upon dissociation from complex I. We previously showed that IKK inhibition induces cytosolic pS166 RIPK1 and complex IIb/necrosome assembly, a set of events that we now found to be prevented by mimicking pS25 RIPK1 (Fig. 2c–g). We also found that IKK inhibition dramatically increases RIPK1 kinase activity in complex I after 10 min of TNF sensing (Fig. 5a), which is in line with the increased autophosphorylation detected by LC-MS/MS already after 5 min of TNF stimulation (Fig. 1c) and that we confirmed by immunoblot (Fig. 5b). Importantly, restoring pS25 RIPK1 with the S25D mutation prevented the boost in pS166 RIPK1 caused by IKK inhibition (Fig. 5c). In line with these results and the ones presented in Figs. 3, 4, TAK1 or SHARPIN inactivation in MEFs, BMDMs or MDFs consistently caused increased RIPK1 kinase activity in complex I, which was also prevented by restoring pS25 RIPK1 (Fig. 5d–f). Altogether, these results therefore suggest that conditions affecting IKK-mediated pS25 RIPK1 in complex I induce a local boost in RIPK1 kinase activity that drives RIPK1 kinase-dependent cell death.

**Phospho-Ser25 directly inhibits RIPK1 catalytic activity.** Serine 25 resides in the kinase domain of RIPK1 within the β1-β2 hairpin and marks the beginning of the Glycine-rich loop in the ATP-binding site (Fig. 6a, b), one of the most highly conserved sequence motifs in protein kinases[50]. This structural motif is the most flexible part of the kinase N-lobe and is well served by this property to seal the bound nucleotide and help positioning the γ-phosphate of ATP for hydrolysis. The kinase domain of RIPK1 has so far only been crystalized in its inactive conformation[51,52], and structural overlays employing all known structures reveal that Ser25 localizes at the tip of this loop and is approximately at the same height as where the phosphate tail of ATP would be bound in the active site (Fig. 6b). Furthermore, Ser25 is roughly halfway between the two ordered halves of the RIPK1 activation loop and would be expected to be adjacent to the disordered part of the activation loop. Interestingly, Ser25 is also present in RIPK2 and in RIPK4 (Fig. 6a) and crystal structures of active RIPK2[53,54] and RIPK4[55] indicate that Ser25 helps to seal the nucleotide binding site near the ribose and α-phosphate of the bound nucleotide (Fig. 6c). Based on these structural considerations, pS25 RIPK1

would be expected to substantially alter the electrostatic properties of the Glycine-rich loop, and its interactions with adjacent parts of the structure, including the activation loop. Thus, pS25 would be expected to impair nucleotide binding due to electrostatic repulsion or RIPK1 kinase activity due to structural perturbation of the activation loop. We interrogated this hypothesis by performing kinase assays using recombinant RIPK1, which showed that the S25D phospho-mimetic mutation greatly affected RIPK1 kinase activity, although not as potently as the K45A kinase-dead mutation (Fig. 6d–e). Phosphorylation on Ser25 therefore appears to function as a physiological mechanism to directly repress RIPK1 catalytic activity during engagement of cell surface receptors.

**The anti-death role of IKKs is not limited to phospho-Ser25.** The results generated so far identified IKKα/β-mediated phosphorylation of RIPK1 on Ser25 as a physiological brake directly repressing RIPK1 kinase activity. However, it remains unclear whether the sole inhibition of this phosphorylation event suffices to activate RIPK1 in complex I and to trigger RIPK1-kinase-dependent death upon single TNF stimulation. To address this question, we reconstituted RIPK1-deficient MEFs with WT RIPK1 or with the S25A phospho-null mutant and stimulated the cells with TNF. We observed that preventing pS25 RIPK1 with the S25A mutation was sufficient neither to activate RIPK1 in TNFR1 complex I nor to trigger cell death in response to single TNF sensing, while additional IKK inhibition similarly activated both readouts in the two cell lines (Fig. 7a, b). The fact that the S25A phospho-null mutation did not mimic IKK inhibition indicates that the repressive role of IKKs on RIPK1 is not limited to Ser25 phosphorylation, but additionally involves phosphorylation of another target, which could either be another residue of RIPK1 or another protein. It is sufficient to mimic one brake on RIPK1 (pSer25) to prevent RIPK1 activation, but all the brakes need to be released to activate RIPK1 and to induce cell death. By combining the S6A and S25A mutations, we excluded the implication of IKK-mediated phosphorylation of RIPK1 on Ser6 (Fig. 7b). In addition, by challenging the S25A reconstituted cells with TNF in presence of CHX, we also excluded genes activated by IKK-dependent phosphorylation (Fig. 7c). Altogether, our results support a two-signal model, whereby IKKs inhibit RIPK1 by repressing its kinase activity through Ser25 phosphorylation and by preventing its activation through phosphorylation of a yet to be discovered additional target (Fig. 7d–e).

**Discussion**
RIPK1 is a complex protein that possesses both a scaffolding pro-survival as well as a catalytic pro-death function. Although dispensable for proper development, RIPK1 kinase-dependent cell death is beneficial and promotes optimal antibacterial immunity

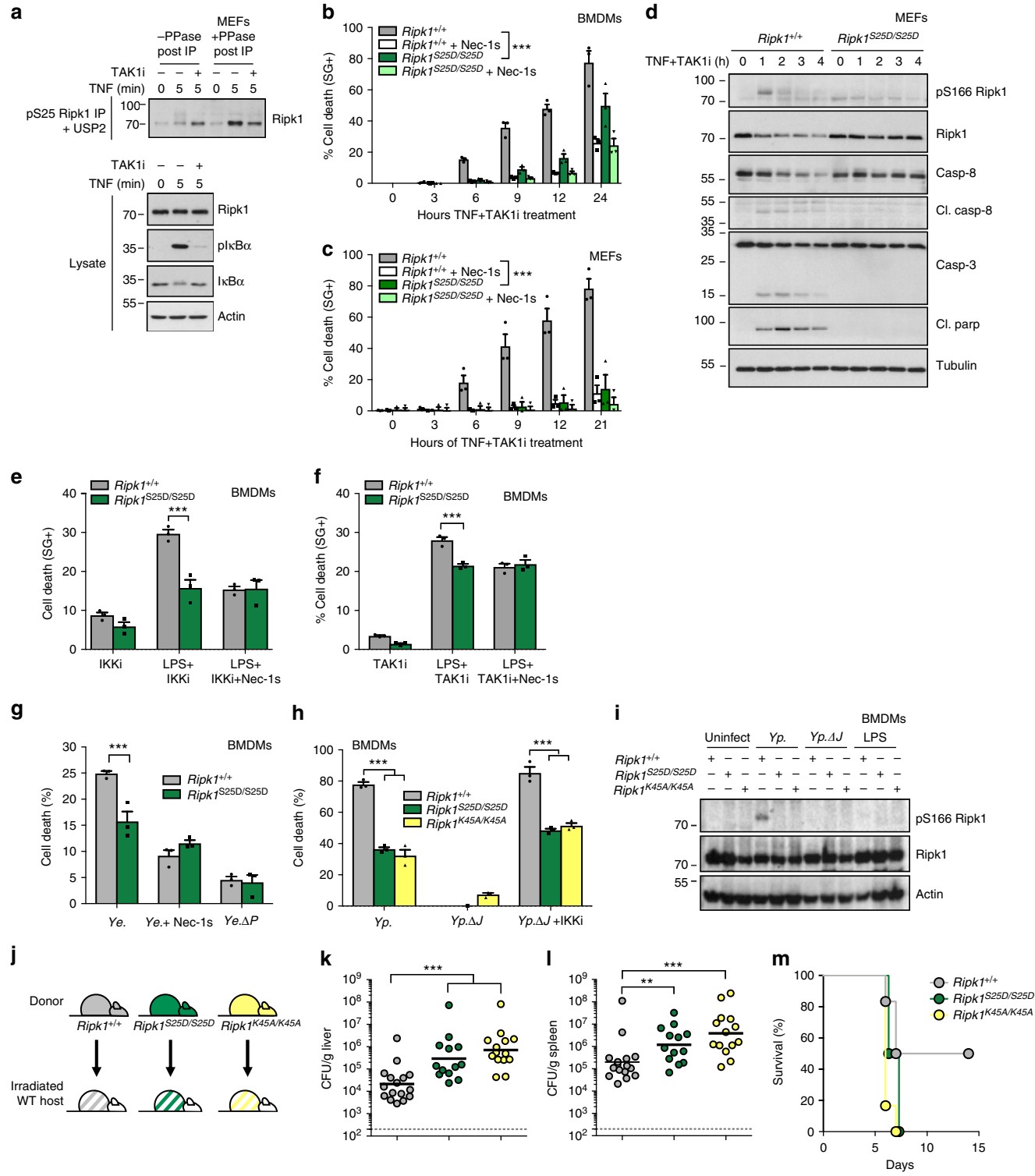

in some infectious models, such as following *Yersinia* infection[6], but is detrimental and drives pathogenesis in various inflammatory diseases, including in the SHARPIN-deficient mice (*Shpn*^*cpdm/cpdm*)[3]. These two examples illustrate the importance of proper RIPK1 enzymatic regulation, but our understanding of the precise molecular mechanism(s) regulating RIPK1 activation/ repression has so far remained limited. Indeed, the development of specific inhibitors and the generation of kinase-dead genetic models have been of great value to identify physio-pathological conditions driven by RIPK1 catalytic activity, but the

physiological mechanisms that regulate the engagement of RIPK1 kinase activity remain poorly defined.

Because no upstream RIPK1-activating kinase has been reported so far, it is currently assumed that catalytic activation of RIPK1 is caused by conformational changes induced upon recruitment of RIPK1 to signaling platforms. Under steady state conditions, binding of RIPK1 to chaperones, such as HSP90[56], or other proteins would maintain RIPK1 in an inactive conformation. The integration of RIPK1 to receptor signaling complexes, such as TNFR1 complex I, would instead favor the adoption of a

**Fig. 3** Phospho-Ser25 regulates the immune response against Yersinia. **a** Wild-type MEFs were pretreated with the indicated compounds for 30 min before stimulation with 1 μg/ml hTNF for the indicated duration. pS25 RIPK1 was then immunoprecipitated and treated with USP2 and λ phosphatase (PPase) post-IP when indicated. **b–f** $Ripk1^{+/+}$ and $Ripk1^{S25D/S25D}$ BMDMs (**b, e–f**) or MEFs (**c, d**) were pretreated with the indicated compounds for 30 min before stimulation with hTNF (10 pg/ml for BMDMs and 100 pg/ml for MEFs) (**b–d**) or 50 ng/ml LPS for 6 h (**e**) or 4 h (**f**). Activation of cytosolic proteins was monitored by immunoblot (**d**) and cell death was measured in function of time (**b–c, e–f**). (**g**) $Ripk1^{+/+}$ and $Ripk1^{S25D/S25D}$ BMDMs were infected with wild-type *Y. enterocolitica* (*Ye*) or with the YopP-negative mutant *Ye.ΔP* in presence or absence of Nec-1s. Cell death was quantified by SytoxGreen staining 4 h post-infection. **h–i** $Ripk1^{+/+}$, $Ripk1^{S25D/S25D}$ and $Ripk1^{K45A/K45A}$ BMDMs were pretreated, or not, for 1 h with 5 μM IKK inhibitor BMS-345541 (IKKi), then infected with wild-type *Y. pseudotuberculosis* (*Yp*) or the YopJ-negative mutant *Yp.ΔJ*. Cell death was measured by LDH 6 h post-infection (**h**) and cytosolic RIPK1 activation was monitored by immunoblotting for pS166 RIPK1 (**i**). Cell death data are presented as mean ± SEM three independent experiments (**c**). BMDMs results were obtained with cells isolated from three ($n = 3$) (**b, e–h**) different mice of each genotype. Statistical significance for the cell death assays was determined using two-way ANOVA followed by a Tukey (**b, c, h**) or Sidak (**g**) post-hoc test. **J** BM chimeras were generated by reconstituting lethally irradiated congenic hosts with BM from either $Ripk1^{+/+}$, $Ripk1^{S25D/S25D}$, or $Ripk1^{K45A/K45A}$ mice. **k–m** BM chimeric mice were infected with 1–2 $10^8$ CFUs *Y. pseudotuberculosis* by oral gavage. Liver (**k**) ($Ripk1^{+/+}$ $n = 16$, $Ripk1^{S25D/S25D}$ $n = 13$, $Ripk1^{K45A/K45A}$ $n = 14$) and spleen (**l**) ($Ripk1^{+/+}$ $n = 16$, $Ripk1^{S25D/S25D}$ $n = 13$, $Ripk1^{K45A/K45A}$ $n = 14$) bacterial burdens were measured on day 5 post-infection and survival was recorded during two weeks (**m**) ($Ripk1^{+/+}$ $n = 6$, $Ripk1^{S25D/S25D}$ $n = 6$, $Ripk1^{K45A/K45A}$ $n = 6$). Statistical significance for the bacterial burdens was determined using a Mann–Whitney test. Significance between samples is indicated in the figures as follows: **$p < 0.01$; ***$p < 0.001$. Immunoblots are representative of two (**d–i**) independent experiments

more active conformation. In such a scenario, post-translational molecular switches within the signaling complex are therefore required to repress RIPK1 catalytic activity in order to prevent uncontrolled cell death induction. Conversely, the switches are turned off in conditions of RIPK1 kinase-dependent cell death. This could occur when RIPK1 kinase-dependent cell death is beneficial for the host, such as following *Yersinia* infection, or in disease conditions caused by genetic mutations, such as the spontaneous mutation that arose in SHARPIN in the chronic proliferative dermatitis mice ($Shpn^{cpdm/cpdm}$). In accordance with this model, we present data demonstrating that Ser25 phosphorylation of RIPK1 by IKKα/β in TNFR1 complex I serves as a physiological molecular brake directly repressing RIPK1 enzymatic activity and, consequently, preventing RIPK1 kinase-dependent complex IIb/necrosome assembly and apoptosis/necroptosis induction. Within complex I, the ubiquitin chains generated by cIAP1/2 and LUBAC (composed of HOIL-1, HOIP and SHARPIN) act as docking stations for TAK1 recruitment and for activation of IKKα/β by TAK1. Accordingly, TNF-mediated pS25 RIPK1 was also affected in SHARPIN or TAK1 deficient cells, and led to catalytic activation of RIPK1 in complex I and RIPK1 kinase-dependent cell death. Restoring RIPK1 phosphorylation in these cells through expression of the S25D phospho-mimetic RIPK1 mutant inhibited RIPK1 kinase activity at the receptor complex and prevented TNF-induced RIPK1 kinase-dependent cell death. In line with these results, mimicking pS25 RIPK1 also protected cells from RIPK1 kinase-dependent cell death induced by YopJ/P, a TAK1 and IKK inhibitor that is injected into host cells by pathogenic *Yersinia* species. RIPK1 kinase-dependent apoptosis of the host hematopoietic cells in response to *Yersinia* infection has been reported to serve as a backup mechanism providing cell-extrinsic signals to promote optimal antibacterial immunity[6]. Accordingly, we found that $Ripk1^{K45A/K45A}$ and $Ripk1^{S25D/S25D}$ chimeras had increased bacterial burden in the liver when compared with WT BM chimeras, which resulted in reduced viability. On the other hand, crossing the $Ripk1^{S25D/S25D}$ mice with the $Shpn^{cpdm/cpdm}$ mice completely rescued their multi-organ inflammatory phenotype, thereby demonstrating the importance of Ser25 phosphorylation in two different models in which RIPK1 kinase-dependent death is, respectively, either beneficial or detrimental for the host. Interestingly, we found that the $Ripk1^{S25D/S25D}$ mice were also protected from the TNF-induced lethal shock caused by RIPK1 kinase-dependent cell death. This suggests that the in vivo inflammatory context caused by TNF injection somehow affects pS25 RIPK1. It was previously reported that co-sensing of TNF with other TNF family ligands, such as TWEAK, switches the TNF response from survival to RIPK1 kinase-dependent apoptosis by affecting cIAP1/2 levels[57,58]. Indeed, we found that the S25D mutation also protected cells from RIPK1 kinase-dependent apoptosis induced by TNF and TWEAK co-stimulation or by TNF stimulation in presence of the cIAP1/2 antagonist BV6 (cIAP1/2i) (Supplementary Fig. 4A–E). In general, we observed that the S25D mutation provided a better protection to the various RIPK1 kinase-dependent cell death triggers in MEFs and MDFs than in BMDMs. This difference may however originate from the in vitro setting since the S25D mutation completely rescued the inflammatory phenotype of the $Shpn^{cpdm/cpdm}$ mice, which originates from the death of cells of various types.

Serine 25 resides in the kinase domain of RIPK1 and is located just next to the Glycine-rich loop, a highly conserved sequence motif within protein kinases that is reported to help coordinating the γ-phosphate of ATP for catalysis. The analysis of all available crystal structures of the catalytic domains of RIPK1, RIPK2, and RIPK4 predicts that Ser25 phosphorylation of RIPK1 would have an important impact on its catalytic activity, either by impairing nucleotide binding due to electrostatic repulsion or by structural perturbation of the activation loop. In accordance with these predictions, we found that the S25D phospho-mimetic mutation greatly affected RIPK1 enzymatic activity in kinase assays performed with purified recombinant proteins. The extent of protection against RIPK1 kinase-dependent cell death was sometimes lower with the S25D mutation then with the K45A kinase-dead mutation or the use of Nec-1s. This may indicate that Ser25 phosphorylation does not fully inhibit but rather greatly represses RIPK1 activity. Alternatively, the absence of full kinase inhibition may result from the fact that a S > D mutation does not fully reconstitute the phosphorylated state of the residue. Of note, the small fraction of cell death not prevented by RIPK1 kinase inhibition in our in vitro cell death assays originates from NF-κB inhibition, as previously reported[22]. The specificity of Ser25 phosphorylation in directly repressing RIPK1 enzymatic activity is also supported by the similarities in the phenotypes of the $Ripk1^{K45A/K45A}$ and $Ripk1^{S25D/S25D}$ mice. Both mouse lines are viable and fertile and only reveal their shared particularities in conditions involving RIPK1 kinase-dependent cell death.

The fact that Ser25 of RIPK1 is only conserved within mammals indicates that other mechanism(s) regulate the catalytic activity of RIPK1 in more evolutionary distant species. Interestingly, RIPK1 might not be the only family member whose catalytic activity is similarly inhibited by phosphorylation. Indeed, both RIPK2 and RIPK4 also possess a Serine residue adjacent to their Glycine-rich loop. In the case of RIPK2, the crystal structure of its catalytic domain bound to the non-hydrolyzable ATP analogue AMP-PCP even indicates that the Ser25 of RIPK2

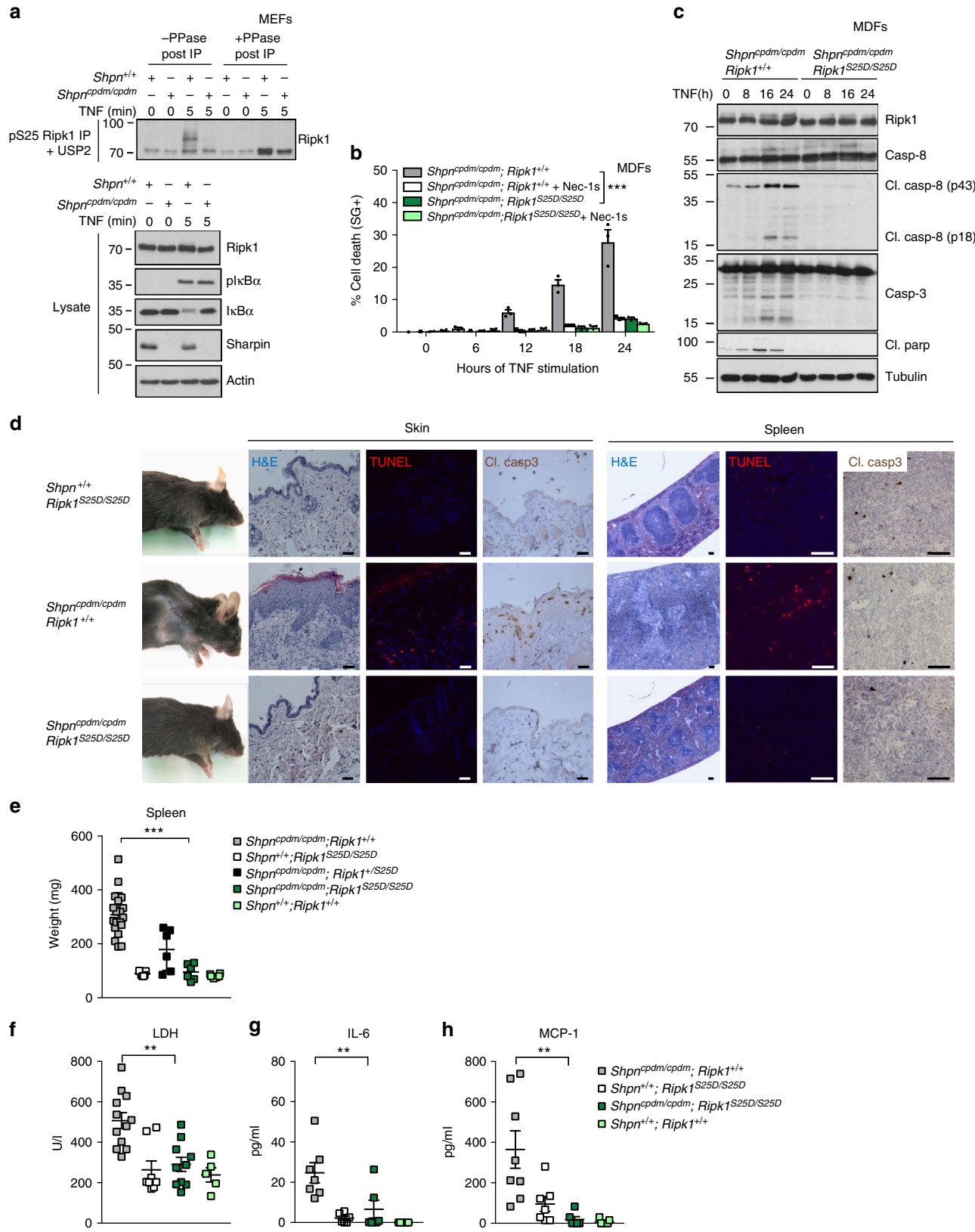

would help coordinating ATP by allowing polar contact[54]. The addition of a phosphate group on this residue is therefore expected to impair nucleotide binding due to electrostatic repulsion.

Our study focused on TNFR1 signaling but RIPK1 also regulates signaling downstream of other immune receptors, including CD95, DR5, TLR-3/4, and RIG-I[1]. Interestingly, despite the fact that all these receptors have the potential to trigger cell death, their default response generally consists in the activation of the NF-κB and MAPKs pathways for the induction of inflammatory mediators. It is therefore tempting to speculate that IKKα/β-mediated phosphorylation of RIPK1 on Ser25 also

**Fig. 4** Defective phospho-Ser25 can drive multi-organ inflammation. **a** $Shpn^{+/+}$ or $Shpn^{cpdm/cpdm}$ MDFs were pretreated with the indicated compounds for 30 min before stimulation with 1 µg/ml hTNF for the indicated duration. pS25 RIPK1 was then immunoprecipitated and treated with USP2 and λ phosphatase (PPase) post-IP when indicated. **b**, **c** $Shpn^{cpdm/cpdm}$; $Ripk1^{+/+}$ or $Shpn^{cpdm/cpdm}$; $Ripk1^{S25D/S25D}$ MDFs were pretreated with the indicated inhibitors for 30 min prior to 1 ng/ml hTNF stimulation. Cell death and protein activation were determined in function of time respectively by SytoxGreen positivity (**b**) and immunoblot (**c**). Cell death data are presented as mean ± SEM of three independent experiments ($n = 3$). Statistical significance for the cell death assays was determined using two-way ANOVA followed by a Tukey post-hoc test. **d** Representative picture of 8-weeks-old $Shpn^{+/+}$ $Ripk1^{S25D/S25D}$, $Shpn^{cpdm/cpdm}$ $Ripk1^{+/+}$, and $Shpn^{cpdm/cpdm}$ $Ripk1^{S25D/S25D}$ mice, and section of their skin and spleen stained for H&E, TUNEL or cleaved caspase-3. Scale bar represents 40 µm. **e** Spleen weight from age-matched 8–12-weeks-old mice ($Shpn^{cpdm/cpdm}$ $Ripk1^{+/+}$ $n = 19$, $Shpn^{+/+}$ $Ripk1^{S25D/S25D}$ $n = 7$, $Shpn^{cpdm/cpdm}$ $Ripk1^{+/S25D}$ $n = 6$, $Shpn^{cpdm/cpdm}$ $Ripk1^{S25D/S25D}$ $n = 6$, $Shpn^{+/+}$ $Ripk1^{+/+}$ $n = 6$). **f** Lactate dehydrogenase (LDH) ($Shpn^{cpdm/cpdm}$ $Ripk1^{+/+}$ $n = 12$, $Shpn^{+/+}$ $Ripk1^{S25D/S25D}$ $n = 8$, $Shpn^{cpdm/cpdm}$ $Ripk1^{S25D/S25D}$ $n = 10$, $Shpn^{+/+}$ $Ripk1^{+/+}$ $n = 5$), **g** IL-6 ($Shpn^{cpdm/cpdm}$ $Ripk1^{+/+}$ $n = 7$, $Shpn^{+/+}$ $Ripk1^{S25D/S25D}$ $n = 6$, $Shpn^{cpdm/cpdm}$ $Ripk1^{S25D/S25D}$ $n = 6$, $Shpn^{+/+}$; $Ripk1^{+/+}$ $n = 7$), and **h** MCP-1 ($Shpn^{cpdm/cpdm}$; $Ripk1^{+/+}$ $n = 8$, $Shpn^{+/+}$; $Ripk1^{S25D/S25D}$ $n = 7$, $Shpn^{cpdm/cpdm}$; $Ripk1^{S25D/S25D}$ $n = 6$, $Shpn^{+/+}$; $Ripk1^{+/+}$ $n = 7$) levels were determined in the serum of 8–12-weeks-old mice. Spleen weight, LDH, IL-6, and MCP-1 are presented as mean ± SEM from n samples from individual mice. Statistical significance was determined by one-way ANOVA followed by a Tukey post-hoc test. Significance between samples is indicated in the figures as follows: $**p < 0.01$; $***p < 0.001$. Immunoblots are representative of two (**a**, **c**) independent experiments

functions as a switch to repress RIPK1 kinase activity downstream of these receptors. Indeed, both TLR4/TRIF and TNFR1 signaling play an important and non-redundant role in RIPK1 kinase-dependent apoptosis in response to *Yersinia* infection[39,59]. RIPK1 kinase-dependent death could have evolved as a general backup mechanism aimed at eliminating cells unable to mount a proper immune response due to IKKα/β inactivation. Phosphorylation of RIPK1 by IKKs on Ser25 therefore appears to act like a guardian of IKKα/β signaling—if IKKα/β signaling is intact, RIPK1 is in an pS25 state, and promotes survival. If IKKα/β or TAK1 are inhibited but TNFR1 or TLR4 signaling is engaged, RIPK1 is in a Ser25-dephospho state, leading to cell death. This would happen physiologically during infection by pathogens that block IKK or TAK1 activation. Further studies will therefore be needed to evaluate the occurrence of Ser25 phosphorylation of RIPK1 by IKKα/β, or alternative kinases, in these additional signaling pathways.

Importantly, our results also demonstrated that the repressive role of IKKs on RIPK1 is not limited to Ser25 phosphorylation. Indeed, the S25A phospho-null RIPK1 mutation did not mimic the effect of IKK inhibition on RIPK1 activation and cell death induction. This implies that IKKs maintain RIPK1 in a pro-survival mode first by repressing its kinase activity through Ser25 phosphorylation and secondly by preventing its activation through phosphorylation of a yet to be discovered additional target. This second target can either be another protein or another residue of RIPK1. Immunoblotting of RIPK1 associated to complex I reveals a phosphorylated smear that collapses in IKKα/β inhibited conditions, suggestive of simultaneous phosphorylation of RIPK1 by IKKα/β on multiple sites[22]. Our LC-MS/MS analysis identified Ser6 as another residue of RIPK1 phosphorylated by IKKs, but our reconstitution experiment with the S6A RIPK1 mutant excluded any implication of this residue in RIPK1 activation. Furthermore, the contribution of IKK-dependent gene activation was also ruled out by challenging the S25A RIPK1 reconstituted cells with TNF in presence of CHX. The identification and characterization of this additional layer of RIPK1 repression by IKKα/β therefore represents an exciting challenge for the future.

Interestingly, p38/MK2 and TBK1/IKKε were also recently reported to phosphorylate RIPK1 to repress its activity[19–21,25–27]. It is however important to mention that only IKKα/β or TBK1/IKKε inhibition, but not p38/MK2 inhibition, suffices to switch the TNF response from survival to RIPK1 kinase-dependent cell death. Those results not only indicate that IKKα/β and TBK1/IKKε control a more critical brake in the TNFR1 death pathway, but also that these kinases cannot substitute for each other in the control of RIPK1. In line with this observation, we found that IKKα/β phosphorylate RIPK1 on Ser25 in a considerable fraction, but not in the entire pool, of RIPK1 associated to complex I. It will therefore also be interesting to evaluate in the future if IKKα/β and TBK1/IKKε phosphorylate identical or different fractions of RIPK1 associated with TNFR1 complex I, especially knowing that Ser25 of RIPK1 was also identified by LC-MS/MS as an IKKε substrate in an in vitro kinase assay[26]. In this study, we reveal that phosphorylation of RIPK1 on Ser25 by IKKα/β serves as a physiological brake that directly inhibits RIPK1 kinase activity, thereby providing insight into the regulation of inflammatory cell death and a potential target for pharmacological intervention. Understanding the precise functional consequences of the other phosphorylation events and their potential to modulate the behavior of RIPK1 in the context of Ser25 post-translational modifications represent exciting future challenges.

## Methods

**Antibodies and reagents**. The following antibodies were used throughout this manuscript: anti-RIPK1 (BD Biosciences #610459, 1:2000; Cell Signaling #3493, 1:2000), anti-pSer32/Ser36 IκBα (Cell Signaling #9246, 1:2000), anti-IκBα (Santa Cruz sc-371, 1:2000), anti-actin (MP Biomedicals #69100, 1:20000), anti-IKKα (Cell Signaling #2682, 1:1000), anti-IKKβ (Cell Signaling #2684, 1:1000), anti-β-tubulin-HRP (Abcam ab21058, 1:10000), anti-pThr180/Tyr182 p38 (Cell Signaling #9211, 1:2000), anti-p38 (Cell Signaling #9212, 1:2000), anti-pThr183/Tyr185 JNK (Invitrogen #44–682 G; 1:1000), anti-JNK (Cell Signaling #9252, 1:1000), anti-pSer166 RIPK1 (Cell Signaling #31122, 1:1000), anti-cleaved caspase-8 (Cell Signaling #9429, 1:1000), anti-caspase-8 (Abnova MAB3429, 1:1000), anti-caspase-3 (Cell Signaling #9662, 1:1000), anti-cleaved PARP (Cell Signaling #9544 S, 1:1000), anti-pSer345 MLKL (Millipore MABC1158, 1:1000), anti-MLKL (Millipore MABC604, 1:1000), anti-FADD (Enzo Life Sciences ADI-AAM-212-E, 1:1000), anti-SHARPIN (Proteintech 14626–1-AP, 1:2000), anti-TRADD (Bio-Rad AHP2533, 1:1000), anti-FLAG-HRP (Sigma-Aldrich A8592, 1:1000), and anti-thiophosphate ester antibody (Epitomics #2686–1, 1:2000). The anti-pSer25 mRIPK1 antibody is a rabbit polyclonal custom-made antibody produced by ThermoFisher Scientific following the 2-rabbit 90-day protocol. Briefly, two rabbits were immunized with 0.25 mg (initial injection) or 0.10 mg (booster injection at day 14, 42, and 56) custom-made phosphorylated peptide (C-LEKTDLD(pS) GGFGKVS-amide). Rabbit serum was collected at day 70 and was purified by subsequent positive selection (affinity towards phosphopeptide) and negative selection steps (no affinity towards non-phosphorylated peptide, C-LEKTDLDSGGFGKVS-amide) to yield the final purified phosphospecific antibody (0.5 mg/ml). Phosphospecificity was further confirmed by ELISA's against the phosphorylated and non-phosphorylated peptides. Recombinant human TNF-α (concentration indicated in the figure legends) and its FLAG-tagged counterpart (1 µg/ml) were purchased from the VIB Protein Service Facility (Ghent, Belgium). LPS (50 ng/ml, Sigma–Aldrich L-2630). Fc-TWEAK was a kind gift of Prof. Dejardin (University of Liege, Belgium) and used at 100 ng/ml. Rat recombinant untagged USP2 was purchased from Enzo Life Sciences (BML-UW9850–0100). Lambda protein phosphatase was obtained from New England Biolabs (P0753s). The following compounds were used: MK2i (PF3644022: 2 µM, Tocris Bioscience), IKKi (TPCA-1: 5 µM, Tocris Bioscience; BMS-345541: 5 µM, EMD-Millipore), zVAD-fmk (50 µM, Bachem), CHX (concentration indicated in the figure legend, Sigma–Aldrich), TAK1i (NP-009245: 1 µM, Analyticon Discovery GmbH; (5Z)−7-Oxozeaenol: 1 µM, Tocris) and cIAP1/2i (BV6: 1 µM, Selleckchem). Nec-1s (UAMC-02197: 10 µM, Laboratory of Medicinal Chemistry, University of Antwerp, Belgium).

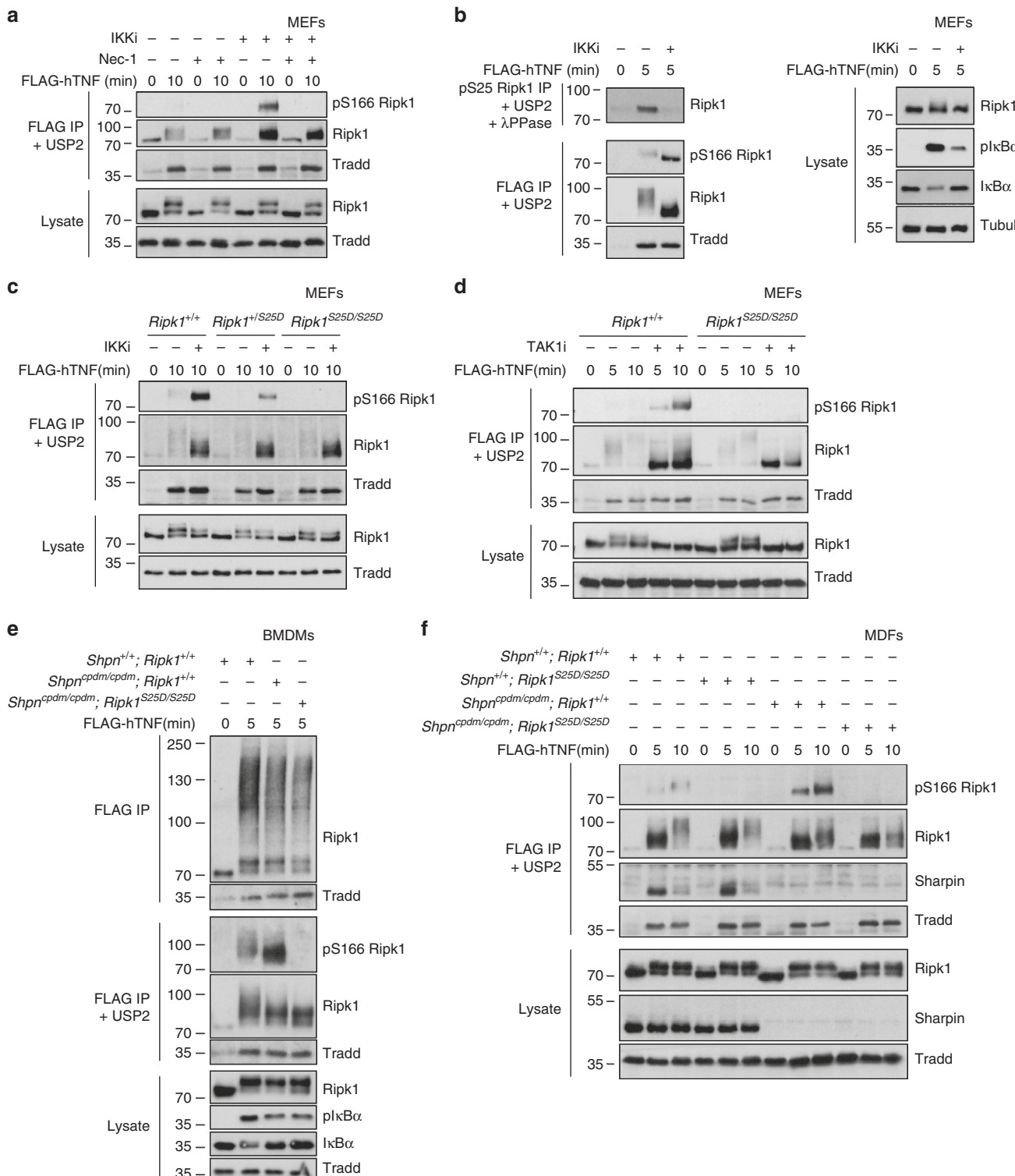

**Fig. 5** Phospho-Ser25 prevents RIPK1 kinase activation in complex I. MEFs (**a–d**), BMDMs (**e**) or MDFs (**f**) of the indicated genotypes were pretreated for 30 min with the indicated compounds before stimulation with 1 μg/ml FLAG-hTNF for the indicated duration. (**a**, **c–f**) TNFR1 complex I was then FLAG-immunoprecipitated and the IPs were then treated with USP2 when indicated. **b** The lysates were then split in two to isolate in parallel TNFR1 complex I by FLAG immunoprecipitation and pS25 RIPK1. The IPs were respectively subsequently treated with USP2 and USP2 + PPase. Protein levels were determined by immunoblot. Immunoblots are representative of two (**a–f**) independent experiments

**Mice.** Generation of the *Ripk1*$^{S25D}$ knockin mouse line by CRISPR/Cas9 was outsourced to Cyagen (USA). The AGC serine-encoding codon in exon 2 of the mouse RIPK1 locus was altered to a GAT codon encoding for glutamate. To do so, Cas9 mRNA, gRNA (5'-gacctagacagcggaggcttcgg-3') generated by in vitro transcription and oligo donor (with targeting sequence, flanked by 120 bp homologous sequences combined on both sides) were co-injected into fertilized C57BL/6 mouse eggs for KI mouse production. F1 heterozygous breeding pairs were obtained from Cyagen and used for establishment of the *Ripk1*$^{S25D}$ colony. *Shpn*$^{cpdm}$ mice were kindly provided by Prof. H. Walczak and were described earlier[44]. Crosses between the *Shpn*$^{cpdm}$ and *Ripk1*$^{S25D}$ were validated by genotyping. All in vivo experiments were performed with littermate mice. Primary cultures were isolated from littermate mice. All experiments on mice were conducted according to institutional,

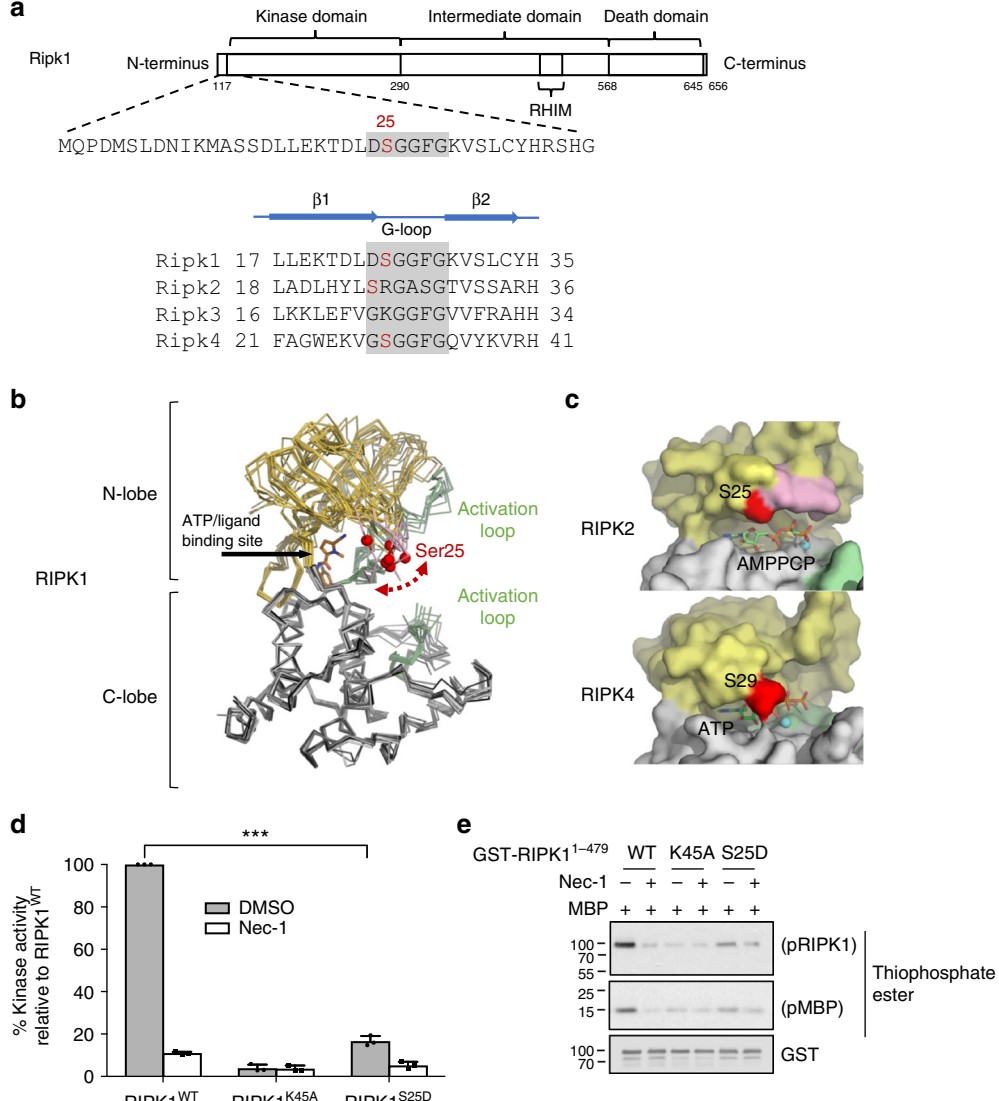

**Fig. 6** Phospho-Ser25 directly inhibits RIPK1 catalytic activity. **a** Schematic representation of mouse RIPK1 primary structure and excerpts from structure-based sequence alignments for murine RIPK1–4 (Uniprot sequences Q60855, P58801, Q9QZL0, Q9ERK0) focusing on the region of Ser25. **b** Ser25 localizes in the flexible Glycine-rich loop covering the RIPK1 nucleotide binding site. Structural overlays were carried out with respect to the C-lobe of RIPK1 in complex with necrostatin-4 (pdb entry 4ITJ) and employed all available RIPK1 kinase domain structures (pdb entries 4ITH, 4ITI, 4NEU, 5HX6, 5TX5, 4M66, 4M69). The red spheres correspond to the C-alpha positions of Ser25 in the different structures. The structurally ordered parts the activation loop are drawn in green. The intervening portion of the activation loop is disordered in all crystal structures of RIPK1 to date. **c** Structures of human RIPK2 in complex with the non-hydrolyzable ATP analog AMP-PCP and RIPK4 in complex with ATP. **d, e** In vitro kinase assays using recombinant truncated RIPK1 mutants (AA 1–479). **d** Quantitative kinase activity measured by ATP consumption using the ADP-Glo kinase assay. Results are presented as a percentage relative to the kinase activity in the wild-type protein and is the mean ± SEM of three independent kinase assays ($n = 3$). **e** Qualitative kinase activity monitored by immunoblot and revealing RIPK1 autophosphorylation and RIPK1-mediated phosphorylation of MBP

national and European animal regulations. Animal protocols were approved by the ethical committee of Ghent University (EC2017–071, EC2017–079).

**Cell lines.** Mouse embryonic fibroblasts (MEFs) and mouse dermal fibroblasts (MDFs) were cultured in Dulbecco's modified Eagle's medium supplemented with 10% fetal calf serum, L-glutamine (200 mM) and sodium pyruvate (400 mM) in normoxic conditions (5% $CO_2$). When the cells were still at the primary state, 0.1% β-mercaptoethanol and penicillin (100 IU/ml) and streptomycin (0.1 mg/ml) was added to the medium. Primary *Ripk1*+/+, *Ripk1*+/S25D, and *Ripk1*S25D/S25D MEFs were isolated from E12.5 littermate embryos following standard protocol and cultured under low-oxygen conditions (3% $O_2$)[60]. MEFs were subsequently immortalized by transfection of a SV40 large T-expressing construct with Jet-PRIME (Polyplus transfection) according the manufacturer's instructions. Wild-type and *Ikkα/β*−/− MEFs were a kind gift of Prof. E. Dejardin and were previously described[22,61]. *Ripk1*+/+ *Shpn*+/+, *Ripk1*+/+ *Shpn*cpdm/cpdm, *Ripk1*S25D/S25D *Shpn*+/+, and *Ripk1*S25D/S25D *Shpn*cpdm/cpdm MDFs were isolated from ears of 8–10-

week-old mice. This was done by sacrificing the mice and subsequently cutting of part of the ear. This piece was then washed in 70% ethanol and the dermis was removed from the ear with forceps. Both the dermis and the leftover part of the ear were then plated in a six-well plate and covered with a sterile microscopy glass cover. When the cells reached confluence (after ~7 days) the cells were passaged and transfected with SV-40 for immortalization. Bone marrow-derived macrophages (BMDMs) were isolated by flushing the femurs and tibias of 7–10-week-old mice with cold PBS using a 26 G needle. The cell suspension was filtered through a 70 μm cell strainer, centrifuged at $220 \times g$, and treated with ACK lysis buffer to remove red blood cells. After centrifugation, the cells were resuspended in RPMI medium supplemented with 10% FCS, 2 mM L-Glutamate, 0.1% β-mercaptoethanol, penicillin (100 IU/ml) and streptomycin (0.1 mg/ml), and 40 ng/ml mouse M-CSF. The cells were plated in a 10 cm non-cell-culture-treated dish and cultured for 7 days at 37 °C in a 10% $CO_2$ incubator. The medium was changed on day 3 and 6. On day 7, the differentiated cells were collected by scraping in cold PBS, centrifuged at $200 \times g$, resuspended in medium without growth factor, and seeded for an experiment.

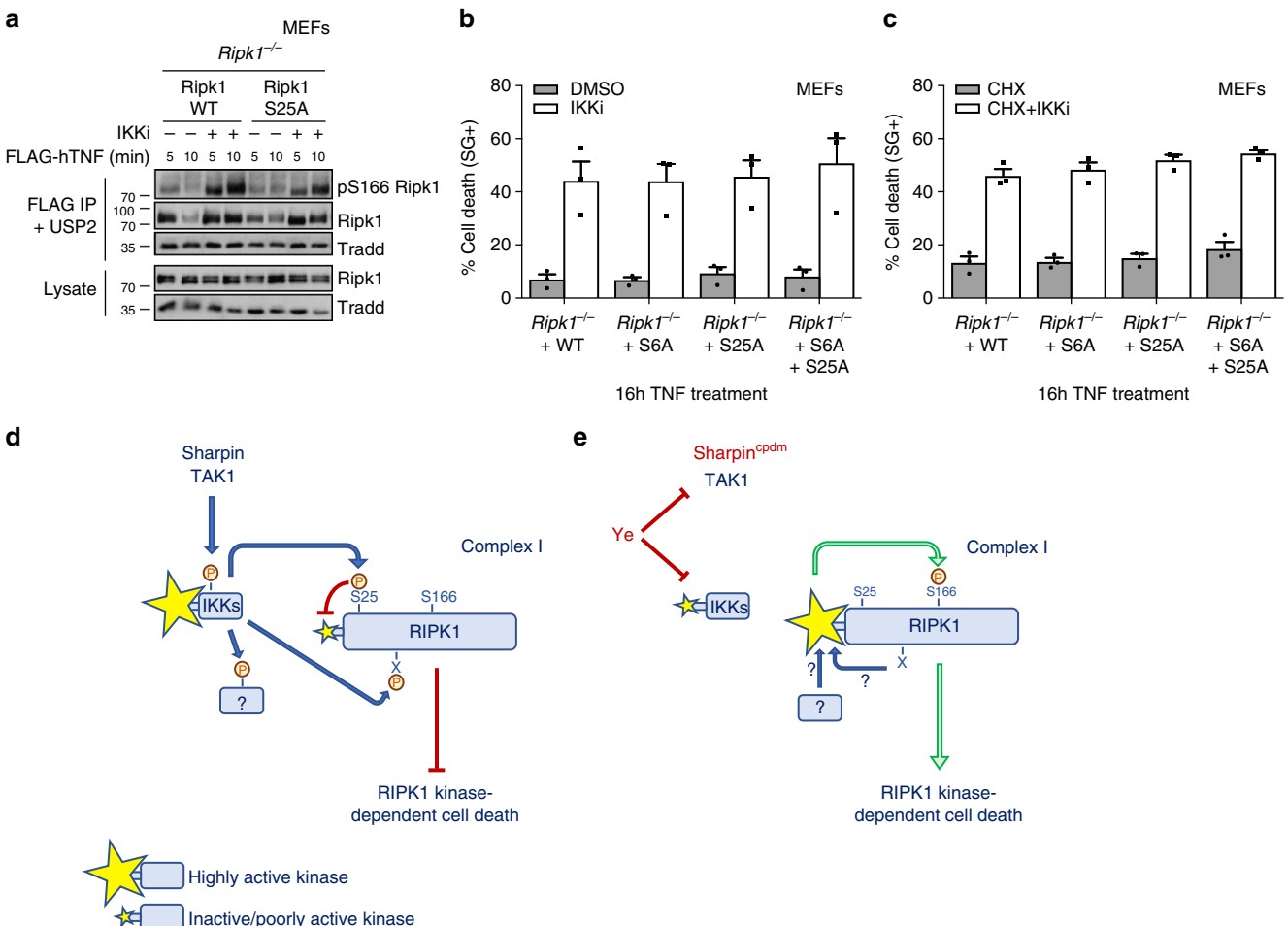

**Fig. 7** The anti-death role of IKKs is not limited to phospho-Ser25. **a–c** $Ripk1^{-/-}$ MEFs were reconstituted with the indicated Ripk1 constructs, pretreated with the indicated compounds for 30 min and stimulated with 1 μg/ml FLAG-hTNF (**a**) or 1 ng/ml hTNF (**b**, **c**). **a** TNFR1 complex I was then FLAG-immunoprecipitated and subsequently treated with USP2. Protein levels were determined by immunoblot. **b**, **c** Cell death was then determined in function of time by SytoxGreen positivity. Cell death data are presented as mean ± SEM three independent experiments. Immunoblots are representative of two independent experiments (**a**). **d**, **e** Two-signal model of IKK-mediated repression of RIPK1, whereby IKKs inhibit RIPK1 first by directly repressing its kinase activity through Ser25 phosphorylation and secondly by preventing its activation through phosphorylation of a yet to be discovered additional target. **d** Under normal conditions, TNF sensing induces the recruitment of RIPK1 to TNFR1 complex I, and the SHARPIN/TAK1-dependent phosphorylation of RIPK1 by IKKs. Phosphorylation on Ser25 by IKKα/β directly represses RIPK1 enzymatic activity and consequently prevents induction of RIPK1 kinase-dependent cell death. **e** In conditions affecting proper IKKα/β activation, such as following *Yersinia* (Ye) infection or in cells bearing the Sharpin^cpdm mutation, defective phosphorylation of RIPK1 by IKKα/β lifts two brakes on RIPK1 kinase activity, resulting in increased autophosphorylation (pS166) and RIPK1 kinase-dependent cell death. Restoring Ser25 phosphorylation (S25D mutation) in these IKKα/β-defective conditions puts one brake back on RIPK1 enzymatic activity, which is sufficient to protect the cell from death. Nevertheless, preventing Ser25 phosphorylation alone is not sufficient to induce RIPK1 kinase-dependent cell death after single TNF ligation, indicative of the existence of an additional substrate of IKKα/β functioning as a second brake on RIPK1 activation

**Immunoprecipitation**. For pSer25 IPs, $7.5 \times 10^6$ cells were seeded per condition in a 150 cm² petri dish. For TNFR1 complex I (CI) and complex II (CII) IPs, $2 \times 10^6$ cells were seeded per condition in a 60 cm² petri dish. The next day, cells were pretreated as indicated in the figure legends and subsequently stimulated (or not) with 1 μg/ml FLAG-hTNF (CI IP), 1 μg/ml hTNF (pS25 IP) or 20 ng/ml hTNF (CII IP). Cells were then washed two times in ice-cold PBS and lysed in 1.2 mL (pS25 IP) or 0.6 mL (CI and CII IPs) NP-40 lysis buffer (10% glycerol, 1% NP-40, 150 mM NaCl and 10 mM Tris-HCl pH 8 supplemented with phosphatase and protease inhibitor cocktail tablets (Roche Diagnostics)). The cell lysates were cleared by centrifugation at $21,000 \times g$ for 15 min at 4 °C and the supernatants were then incubated overnight with FLAG M2 affinity gel (Sigma–Aldrich) for the CI IPs, with protein G Sepharose 4FF (GE Healthcare) + 25 μg anti-pSer25 RIPK1 antibody for the pS25 IPs or with protein G Sepharose 4FF (GE Healthcare) + 3 μL homemade polyclonal rabbit anti-mouse caspase-8 for the CII IPs. The next day, the beads were washed three times in NP-40 lysis buffer. Beads were then resuspended in 60 μL 1x laemmli buffer for direct analysis. When indicated, protein complexes were additionally deubiquitylated (by USP2 treatment) and dephosphorylated (by lambda protein phosphatase treatment) post-IP. To do so, beads were resuspended after the final wash step in 50 μL 1x DUB/λPP buffer (50 mM

Tris-HCl pH 8, 50 mM NaCl, 5 mM DTT and 1 mM MnCl₂). Subsequently, 1.2 μg USP2 (Enzo Life Sciences) and 400U λ PPase were added when indicated. Enzymatic reactions were allowed to proceed for 30 min at 37 °C and subsequently quenched by the addition of 12.5 μL 5x laemmli buffer. For the two successive IPs in Fig. 1i-l, 1% SDS was used to release and dissociate the first complex. The eluate was then diluted 1/10 in NP-40 buffer prior to the second IP. All IPs were analyzed by immunoblotting. Uncropped blots can be found in Supplementary Figure 5.

**Mass spectrometric analysis of phospho-peptides**. $5 \times 10^6$ WT or $Ikk\alpha/\beta^{-/-}$ MEFs were seeded in ten 150 cm² petri dishes per condition. The next day, cells were stimulated for 5 min with 2 μg/ml FLAG-hTNF and then immunoprecipitated according to the CI IP protocol (see immunoprecipitation section). After the final wash step in NP-40 lysis buffer, the beads were additionally washed five times in ice-cold PBS. Protein complexes were then eluted from the beads by addition of 200 μL 150 μg/ml 3x FLAG-peptide (Sigma-Aldrich) in PBS, followed by incubation for 30 min at 22 °C. The eluate was then separated from the beads by applying the samples over Mobicol (MoBiTec) columns with a 35 μM pore size. Proteins were then reduced, alkylated and digested using the FASP protocol. Next,

phospho-peptides were enriched by using Ti$^{4+}$-IMAC before LC-MS/MS analysis. Raw mass spectrometric data were processed with MaxQuant version 1.5.3.28. See Supplemental Information for a more detailed description of the mass spectrometry procedures. All raw mass spectrometry data files and MaxQuant output files have been deposited to the ProteomeXchange Consortium (http://proteomecentral.proteomexchange.org) via the PRIDE partner repository[62] with the dataset identifier PXD009857.

**Histology and analysis of serum markers.** For histology, organs were fixed in 4% paraformaldehyde, embedded in paraffin and cut at 5 μm thickness. Subsequently, sections were stained with hematoxylin and eosin. TUNEL assay was performed according to manufacturer's instructions (In situ cell death detection kit, TMR red—Roche). For immunohistochemistry, sections were deparaffinized, rehydrated and antigen retrieval was done with a citrate buffer (Vector Laboratories). Sections were then incubated overnight at 4 °C with a Caspase-3 primary antibody (Cell Signaling #9661 S, 1:300) followed by a biotinylated secondary antibody (DAKO) and ABC (Vector Laboratories). Serum LDH levels were obtained at UZ-Gent (Belgium) using Cobas 8000 modular analyzer series (Roche Diagnostics, Basel, Switzerland). Serum cytokines levels were measured using a Bio-Plex Multiplex immunoassay (Bio-Rad #171304070) according to the manufacturer's instructions.

**TNF shock model.** For the TNF shock model, 10 μg/20g mTNF (diluted in endotoxin-free PBS pH6.8) was intravenously (i.v.) injected. For the IKKi-sensitized shock model, DMSO stock solutions of IKKi (TPCA-1) were diluted in endotoxin-free PBS pH 6.8, in the presence of 3% Cremophor A25, and injected i.v. in a volume of 200 μl (6% DMSO final) at 10 mg/kg (21.6 μmol/kg) 15 min before i.v. injection of mTNF (150 ng/20 g in 200 μl endotoxin-free PBS pH6.8). Control mice received an equal amount (i.v.) of DMSO (6%) dissolved in DPBS (Vehicle) 15 min before the mTNF challenge. Mortality and body temperature were monitored until 3 days after mTNF injection. Rectal body temperature was recorded with an industrial electric thermometer (Comark Electronics, Norwich, UK; model 2001). Dead mice were considered to be at 25 °C.

**Yersinia infections.** *Y. enterocolitica* in vitro infections were performed with the serotype O8 wild-type strain WA-314 and its isogenic *YopP*-knockout mutant *Ye. ΔP*[63]. Prior to infection, bacterial overnight cultures grown at 27 °C were diluted 1:10 in fresh Luria-Bertani broth and grown for another 1.5 h at 37 °C to activate the *Yersinia* Yop secretion machinery. Cells were infected at a ratio of 20 bacteria per cell and gentamicin (50 μg/ml) was added 90 min after onset of infection to prevent bacterial overgrowth. Infections with WT and isogenic *YopJ*-knockout mutant *Yersinia pseudotuberculosis* strains (*Yp* and *YpΔJ* IP2666 strain background) were performed as previously described[59]. Briefly, *Yp* strains were grown overnight in 2x yeast-tryptone broth shaking culture overnight at 26 °C. Bacterial cultures were diluted the next day into 2xYT containing 20 mM sodium oxalate and 20 mM MgCl2, and grown for an additional 2 h at 26 °C followed by 1 h at 37 °C to induce Yop expression and secretion. Bacteria were harvested, washed three3 times in pre-warmed DMEM, and used to infect BMDMs at 20 MOI.

For in vivo *Yersinia pseudotuberculosis* infections, BM chimeras and infections were performed as previously published[6], with the difference that $3–5 × 10^6$ frozen BM cells were used per mouse to reconstitute. The CFU for oral infection were $1–2 × 10^8$ CFU/mouse by oral gavage.

**Cell death assays.** Cells (MEFs:10,000 per well, BMDMs: 30,000 per well) were seeded in duplicates or triplicates the day before in a 96-well plate. The next day, cells were pretreated with the indicated compounds for 30 min and then stimulated with the indicated concentration of hTNF in the presence of 5μM SytoxGreen (Invitrogen) and 20 μM Ac-DEVD-MCA (PeptaNova). SytoxGreen intensity and caspase-3 activation was measured at intervals of 1 h using a Fluostar Omega fluorescence plate reader, with an excitation filter of 485 nm (SytoxGreen) or 360 nm (Ac-DEVD-MCA), an emission filter of 520 nm (SytoxGreen) or 460 nm (Ac-DEVD-MCA), gains set at 1100, 20 flashes per well and orbital averaging with a diameter of 3 mm. Percentage of cell death was calculated as (induced fluorescence-background fluorescence)/(max fluorescence-background fluorescence)∗100. The maximal fluorescence is obtained by full permeabilization of the cells by using Triton x-100 at a final concentration of 0.1%. All cell death and caspase-3 activation data are presented as mean ± SEM of n (indicated in the Figure) independent experiments, unless stated otherwise. Cell death measured by lactate dehydrogenase (LDH) release was performed using the Cytotox96 Assay kit (Promega) according to manufacturer's instruction.

**Production and purification of recombinant RIPK1.** Recombinant human WT and mutants RIPK1 (aa 1–479) were produced in Sf9 insect cells as GST-fusion protein. hRIPK1 WT GST-fusion construct was obtained by cloning hRIPK1 WT cDNA into the BamHI restriction site of the pAcGHLT vector (BD Biosciences), subsequently hRIPK1 mutants were generated by quickchange mutagenesis. Recombinant baculovirus was obtained after co-transfection of these constructs with ProEasy linearized baculovirus (AB Vector) into Sf9 cells according to the manufacturer's instructions. Sf9-cell pellets were resuspended in 20 mM Tris-HCl

pH 8.0, 200 mM NaCl, 1 mM EDTA, 0.5% (v/v) Igepal CA-630, EDTA-free Protease Inhibitor Cocktail Tablets (Roche Diagnostics). Lysates were incubated on ice for 30 min. Insoluble proteins were removed by centrifugation at $10000 × g$ for 15 min. The supernatant was applied to a Glutathione Sepharose 4FFcolumn (GE Healthcare) pre-equilibrated with PBS pH 7.4. The GST-tagged RIP kinase was eluted from the column with 50 mM Tris-HCl pH 8.0, 100 mM NaCl, 15 mM reduced glutathione. Fractions containing RIPK1 were pooled and further purified using a Superdex 75 pg column (GE Healthcare, running buffer: 20 mM Tris-HCl pH 8.0, 100 mM NaCl). The purity of the fractions was checked by means of SDS-PAGE, the RIPK1 fractions were pooled. 10% glycerol and 5 mM DTT were added to the protein fraction, followed by storage at −70 °C.

**Kinase assays.** Quantitative in vitro kinase assays were performed by using the ADP-Glo kinase assay kit (Promega). In brief, the different recombinant hRIPK1 were incubated at 150 nM for 4 h at room temperature in kinase assay buffer (50 μm ATP, 25 mM HEPES pH 7.5, 25 mM NaCl, 15 mM MgCl2, 0.25 mg/ml BSA, 0.01% CHAPS and 2 mM DTT). To convert ATP consumption into light production, a 2:2:1 (kinase assay reaction:ADP-Glo reagent:kinase detection reagent) ratio of the kit's components was used. Luminescence was measured during 1 s reads with the GloMax 96 microplate luminometer (Promega). For the kinase assays detected by immunoblot, the different recombinant hRIPK1 were incubated with myelin basic protein (Sigma, Cat. No M1891) in kinase assay buffer (166 μm ATPγS (Sigma; Cat. No A1388), 20 mM HEPES pH 7.5, 10 mM MgCl2, and 2 mM DTT). Subsequently, proteins were alkylated with p-nitrobenzyl mesylate (PNBM) (Abcam ab138910) for 90 min at room temperature and analyzed by immunoblotting.

**Reconstitution of RIPK1-deficient cells.** The sequences encoding WT RIPK1 and the mutated versions of RIPK1 were cloned into pENTR3C using the cloneEZ PCR cloning kit (GenScript). These sequences were then recombined into the pLenti6-V5-blasticidin destination vector (for the phospho-mimetic mutants) or the pSIN-3xHA-TRE-GW (for the phospho-null mutants) using the LR gateway recombination system (Invitrogen). Lentiviral reconstitution of MEFs and Jurkats was done by standard protocol. Briefly, HEK293T cells were transfected using calcium phosphate with the plasmids containing the different versions of RIPK1 in combination with the lentiviral packaging vectors pMD2-VSVG and pCMV-ΔR8.91. The medium was changed after 6 h, and collected 48 h post-transfection. The virus-containing supernatant was then used to infect MEFs or Jurkats. Forty-eight hours after infection, plasmid-containing cells were selected (10 μg/ml blasticidin for pLenti6 containing cells and 2 μg/ml puromycin for pSIN containing cells).

**Reporting summary.** Further information on experimental design is available in the Nature Research Reporting Summary linked to this article.

## Data availability

All raw mass spectrometry data files and MaxQuant output files have been deposited to the ProteomeXchange Consortium (http://proteomecentral.proteomexchange.org) via the PRIDE partner repository with the dataset identifier PXD009857. Source data for all graphs can be found in the source data graphs file. Uncropped blots can be found in Supplementary Figure 5.

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

## Acknowledgements

We are grateful to K. Lemeire (VIB-UGent, Belgium), T. Divert (VIB-UGent, Belgium), B. Gilbert (VIB-UGent, Belgium) and S. Men Choi (VIB-UGent, Belgium) for technical assistance. We would like to thank Prof. A. Degterev (Tufts University, USA) and Dr. R. Merceron (VIB-UGent, Belgium) for scientific advices. We also thank Prof. H. Walczak (UCL, UK) for the *Shpn$^{cpdm}$* mice, Dr. J. Bertin (GSK, USA) for the *Ripk1$^{K45A}$* mice, Prof. J. Silke (WEHI, Australia) for the *Shpn$^{cpdm}$* MDFs and Prof. E. Dejardin (GIGA, Belgium) for the *Ikkα/β$^{−/−}$* MEFs. Research in the group of Prof. M.J.M. Bertrand is financially supported by the Vlaams Instituut voor Bio-technologie (VIB)(Tech Watch co-funding), by the Ghent University, by grants from the Fonds voor Wetenschappelijk Onderzoek Vlaanderen (FWO)(G013715N, G044518N, EOS MODEL-IDI 30826052), from the Belgian science policy office (BELSPO)(IAP 7/32) and from the Flemish Government—accorded to Prof. P. Vandenabeele (Methusalem

BOF09/01M00709 and BOF16/MET_V/007). Dr. Y. Dondelinger is supported by a post-doctoral fellowship from the FWO. T. Delanghe and D. Priem have a strategic basic research PhD fellowship from the FWO. Dr. D. Rojas-Rivera. was paid by FWO grant G013715N. The proteomics research in the group of Prof. A. Heck was financially supported by the Netherlands Organization for Scientific Research (NWO) through funding of the large-scale proteomics facility Proteins@Work (project 184.032.201) embedded in the Netherlands Proteomics Centre. Prof. K. Ruckdeschel obtained funding by the Deutsche Forschungsgemeinschaft.

## Author contributions

M.J.M.B. supervised the study. Y.D., T.D., D.P. and M.J.M.B. designed, performed, and analyzed most of the experiments. D.R-R. and T.D. performed and analyzed the TNF-induced shock experiment. R.R. and P.V. designed, performed, and analyzed the kinase assays performed with the purified recombinant RIPK1 proteins. P.G. and A.H. designed, performed, and analyzed the mass spectrometry experiment. M.A.W., D.H.S. and I.B. designed, performed, and analyzed the *Yersinia pseudotuberculosis* infection experiments. J.G. and K.R. designed, performed, and analyzed the *Yersinia enterocolitica* infection experiment. S.N.S. carried out structural analysis of RIPK crystal structures and generated prediction models for Ser25 phosphorylation. M.J.M.B. and Y.D. wrote the manuscript, and the co-authors provided feedback on the text.

## Additional information

**Competing interests:** The authors declare no competing interests.

