## [Peer Review File · Nature Communications]

Reviewers' comments:

Reviewer #1 (Remarks to the Author):

In this paper, Dondelinger et al. investigated on the function of Ser25 (S25) phosphorylation on RIPK1. The authors first identified RIPK1 is phosphorylated at S25 in response to TNF stimulation in TNFR1-associated inflammatory complex I. They showed S25 phosphorylation is dependent on IKK γ . A phospho-mimic mutation of S25 (S25D) greatly reduced RIPK1's ability to mediate TNF-induced RIPK1 kinase dependent apoptosis and necroptosis and protected mice from TNF shock. The authors further promoted S25 phosphorylation directly inhibits RIPK1's kinase activity by comparing WT and S25D RIPK1 kinase activity both in in vitro kinase assays and in in situ TNF-induced Complex I pull-downs. In line with S25D mutant's ability to inhibit RIPK1 kinase activity, RIPK1 S25D mutant mice were shown to be worse in controlling *Yersinia* infections and crossing with RIPK1S25D/S25D mice was shown to rescue Sharpincpdm/cpdm (cpdm) mice's multi-organ inflammatory phenotype.

Overall, this is an interesting and well-written study that would be of interest to the field in that it defined the role of a novel phosphorylation site on RIPK1 to inhibit RIPK1 kinase activity and prevent RIPK1-mediated cell death. The mechanistic studies are backed up with in vivo mouse models (*Yersinia* infection and cpdm mice), demonstrating the significant impact of this modification. In general conclusions are well backed up but there are certain aspects of the manuscript that in this reviewer's opinion should be addressed.

Main points.

1. Identification of IKK-dependent RIPK1 phosphorylation.

The mass spectrometry data depicted in Fig 1C is largely uninformative as only one "dot" is labelled. Please indicate the identification of more relevant phospho-sites and include the original MS data in supplemental data. Also, it would be instructive if the authors could include a table/figure indicating a) all the RIPK1 phospho-sites identified and b) the frequency of Phos-S25 peptides versus unmodified S25 peptides.

The relative occupancy of the S25 phosphorylation, both in RIPK1 associated with TNFR1 complex I and in cytosol, is particularly important for the conclusion that IKK-mediated phosphorylation is an important checkpoint to prevent RIPK1 activation as this would require that a substantial proportion of the RIPK1 molecules within complex I are modified on this site. This would also be a relevant consideration when using S25D knockin cells where all the RIPK1 are mutated.

Related to this, it is somewhat surprising that mice heterozygous for the S25D mutation are almost completely rescued in the in vivo TNF models (Fig. 2H-K). As it implies that the S25D mutation dominantly represses activation of WT RIPK1. It would be of interest to know if MEFs or BMDMs isolated from these mice are protected from TNF-induced cell death (in presence of IKKi +/- zVAD) and fail to activate RIPK1. If indeed the S25D mutation is dominant, in vitro kinase assays could be done with mix of WT and S25D RIPK1 to investigate if S25D mutant will prevent kinase activity of WT RIPK1.

2. IKKs directly phosphorylate RIPK1 S25.

The conclusion of IKKs directly phosphorylate RIPK1 on S25 is drawn from a previous report, where S25 was one of the hit sites from a Mass Spectrometry analysis of RIPK1 phosphorylation sites by IKKs (Fig. S4C of Ref 22 of the paper). As Mass Spectrometry experiments could have false positive hits, an in vitro kinase assay or its similarity should be performed for validation before concluding that IKKs directly phosphorylate RIPK1.

3. The function of RIPK1 S25 phosphorylation.

All experiments regarding the function of S25 phosphorylation is based on S25D phospho-mimic

mutant. It is possible that the phospho-mimic mutant alters protein function to reduce its kinase ability in a way not related to the mimicking of phosphorylation. To substantiate the functional importance of S25 phosphorylation, cells reconstituted with the RIPK1 S25A mutant should also be tested as it would be expected that they display increased cell death compared to WT. It would also be important to demonstrate *in vitro* that S25-phosphorylated RIPK1 activity is inhibited, and its inhibition level compared to RIPK1 S25D mutant.

In addition, the authors claim that RIPK1 S25D protects from TNF-induced cell death but the assays, in particular when assessing MEFs, are short term assays that are ended within 24 h and in some cases only 7 h (e.g. Fig 2E). Thus, it is not clear if the S25D mutation provides long-term protection from cell death or if it just delays cell death. For example, the time-dependent cell death rates shown in Figure 3B-C suggests that cell death in the S25D cells is simply delayed and would reach the same level as in WT cells if the assays was extended. Yet, the authors state that the S25D mutation "greatly protects from cell death induced by TNF when TAK1 is inhibited". It would be appropriate (and more accurate) to specify in the results section and discussion if the mutation provides long-term protection or delays cell death.

4. S25 phosphorylation directly inhibits RIPK1 activity

All the cell death experiments are performed in mouse cells but the *in vitro* analysis of the effect of the S25D mutation on RIPK1 is performed using human RIPK1. Since the necroptosis pathway is relatively unconserved evolutionarily, the *in vitro* experiments should be performed with mouse RIPK1.

Minor comments

1. Recently, Xu et al. reported phosphorylation on T189 of RIPK1 by TBK1 in TNFR1-associated inflammatory complex (Xu, D. et al. (2018). TBK1 Suppresses RIPK1-Driven Apoptosis and Inflammation during Development and in Aging. *Cell*, 174(6), 1477-1491.). T189 phosphorylation site was reported to also directly inhibit RIPK1's kinase activity. It would be relevant to discuss this and comment on the relation between the two phosphorylation sites that both directly inhibit RIPK1 kinase activity.

2. Line 82: The list of proteins in complex I is not complete; TAB-TAK1, the IKK complex, A20, CYLD, Spata2 also have been found in TNF pulldown experiments.

3. Line 127: The authors state "defects in this IKK / checkpoints presumably also explain the *in vivo* inflammatory phenotypes caused by..., such as in NEMO-deficient mice, SHARPIN-deficient mice or mice in which TAK1/IKKs are inhibited following *Yersinia* infection". It should be noted that impaired NF- κ B-mediated transcriptional response also contributes to excessive cell death, and therefore may contribute to the phenotype of the mice.

4. Line 164: The authors conclude the specificity of the protective effect of the S25D mutation only under conditions affecting IKK activation by comparing cell death percentage of WT and RIPK1 S25D cells under TNF+IKKi and TNF+CHX conditions. However, the difference between the two conditions could also be explained by the requirement of RIPK1 kinase activity only in TNF+IKKi-induced cell death (as indicated by Nec-1s inhibition).

5. Line 170: Fig. 3J-K should probably be Fig. 2J-K.

6. In figure 3F, TAK1i is written as TAKi.

7. Lines 284 and 303, SHARPIN-deficient mice are referred to as Shp cpdm/cpdm, while in all other places they are referred to as Shpn cpdm/cpdm.

8. Lines 411 and 413, compound concentrations are missing for BMS-345541 and (5Z)-7-Oxozeaenol.

9. Throughout the paper, certain numbers are not expressed clearly. For example, 7.5.106 (line 450) should be 7.5 106, etc. There is also a mixture of using comma and full stops as decimal points, e.g. lines 467 and 469.

Reviewer #2 (Remarks to the Author):

The manuscript by Dondelinger et al. identifies the target of IKK kinases on RIPK1 as Ser25 residue, located next to the P-loop of the kinase. This phosphorylation event inhibits catalytic activity of RIPK1, thus directing the signaling away from cell death responses. The authors go on to show that phospho-mimetic S25D mutant of RIPK1 indeed lacks pro-death activity in a variety of in vivo scenarios, including two different TNF-induced shock models, the model of *Yersenia* infection, and a model of TNF-dependent inflammatory pathology in mutant Sharpin mice. The critical role of RIPK1 catalytic activity in all of these paradigms is already established, thus new data serves as a confirmation of S25 site serving as a critical element of RIPK1 regulation during inflammation and infection. Overall, I find that this is an excellent and timely story which further elucidates a critical event in the regulation of TNF/RIPK1-dependent cell death. All the experiments are performed at a high technical level. I also believe this story will attract major interest. I just have a few relatively minor comments:

1. The authors should present the phospho-peptide data from Fig. 1C in a Table format. It would appear that multiple phosphorylation events were detected, but it is difficult to evaluate mass spec data using just the presented plot.

2. S25D mutant of RIPK1 should be treated with some caution. It is clearly a kinase-inactive version of RIPK1 protein, thus its lack of function in multiple assays is expected. It certainly likely mimics a phosphorylated state of RIPK1, however, there is still some chance that the observed functional changes specifically reflect D25 residue. Thus, two critical experiments should be added with an opposite S25A mutant of RIPK1. Namely, the authors should confirm that S25A mutant shows S166 phosphorylation after stimulation with TNF alone, similar to the effect of IKKi. Further, the authors should examine whether S25A mutant renders cells sensitive to TNF alone (or perhaps to TNF in combination with MK2i).

3. MK2-dependent phosphorylation of the Ser320 cluster on RIPK1 was previously reported to be critical for the survival of *Yersenia*-infected cells (PMID: 28920954), similar to Ser25 role. Thus, it would appear that both Ser25 and Ser320 sites ultimately function to limit the pro-death activity of RIPK1 downstream from TAK1/IKKs. It is possible that Ser25 is indeed the major site, and Ser320 cluster plays a further modulatory role. I would suggest that the authors should test a triple D mutant of S25/S321/S336 to see if this renders RIPK1 completely inactive in TNF-induced pro-death assays as opposed to some residual activity seen for S25D (for instance, in Fig. 3).

4) The authors should check whether S25D mutation has any effect on pIKK/I κ B degradation following TNF stimulation.

Reviewer #3 (Remarks to the Author):

RIPK1 is a very important regulatory protein involved in key biological processes including cell death and inflammation. It is therefore an interesting drug target for the treatment of disorders such as chronic inflammatory diseases and cancer. In this manuscript, Dondelinger et al. investigate the role of phosphorylation on RIPK1 activities. The authors had previously demonstrated that RIPK1 serine 25 was directly phosphorylated by IKK- α / β in vitro. Now

using a comparative mass spectrometric approach, they confirm that IKK-alpha/beta phosphorylates RIPK1. They demonstrate by an appropriate series of experiments that serine 25 phosphorylation is a major mechanism of inhibition of RIPK1 kinase activity, which prevents TNF-mediated cell death in murine IKK, TAK-1 and SHARPIN deficient murine cells. In addition they extend their findings to several mouse models in vivo. They expectedly show that phosphorylation of RIPK1 on serine 25 leads to a defective cell death-dependent control of *Yersinia enterocolitica* and *Yersinia pseudotuberculosis* infections, to an improved control of inflammation in SHARPIN deficient mice and to a better survival of mice treated with IKK inhibitors.

Major comments

- 1-The experiments were performed in mice and murine cells. The study would benefit from a comparison of the findings in human and murine cells, RIPK1 being a critical regulator of immunity in humans.
- 2-Most figures show that pSer25 does not fully prevent cell death. The pSer25-independent mechanisms of protection should be discussed more thoroughly.
- 3-A control of IKK inhibition levels upon treatment of mice with IKK inhibitors should be provided.
- 4-Add a comment on the relative importance of the cell death inhibitory role of pSer25 depending on the cell type, the mouse line etc. (Figure 2B, figure 3C, figure 4B).
- 5-Add a statistical analysis of the results shown in figure 6D.

Other comments

Line 189: Gram

Line 300: change intentionally/unintentionally

Line 342: almost as potent is too strong

Line 432: 0.1%

Lines 432 and 444: the concentration of P/S is very low

Lines 434 and 439: add references for these protocols

Line 501: add the protocol identification number

Discussion: a model that could be Figure 7 or Supp figure S5 would be helpful

Figure 1G: explain the surprising profiles of the I κ B and pI κ B blots (TNF 10 and 20 min)

Figure 3K: the increase in CFU of RIPK1S25D mice is quite small. It would be interesting to measure bacterial burden in other organs.

Figure 6A: add sizes of the domains and the size of full length RIPK1

Figure 6E: Thiophosphate

Figure S1: low levels of RIPK1 are surprisingly detected in the RIPK1S25A mutant after pS25 immunoprecipitation. How was the specificity of the antibody tested?

We would first like to thank the editor and the reviewers for their enthusiasm regarding our study and for their insightful comments. We performed numerous experiments to address the points raised by the reviewers and have extensively adapted our manuscript accordingly (more than 15 new panels). The modifications appear in red in the text and Figures). We would also like to apologize for the non-accessibility to the full mass spectrometry data in the first submission. A link giving access to the results had been provided in the cover letter and reporting summary, but it was probably not at the right place since the information was not transmitted to the reviewers. Please find below our point-by-point answers to the reviewers' comments.

Reviewer #1:

Main points:

1. Identification of IKK-dependent RIPK1 phosphorylation.

The mass spectrometry data depicted in Fig 1C is largely uninformative as only one "dot" is labelled. Please indicate the identification of more relevant phospho-sites and include the original MS data in supplemental data. Also, it would be instructive if the authors could include a table/figure indicating a) all the RIPK1 phospho-sites identified and b) the frequency of Phos-S25 peptides versus unmodified S25 peptides.

- As mentioned above, a link giving access to the full mass spectrometry data had been included in the cover letter, but this info was unfortunately not transmitted to the reviewer. We apologize for that and now provide the full list as supplementary Table 1. We agree with the reviewer that the original Fig 1C was not very informative. As suggested, we replaced this panel by a table indicating all identified RIPK1 phosphosites by LC-MS/MS, and the ratio of their abundance between WT and IKK α/β DKO MEFs.

Fig. 1C

	Residue	Log ₂ WT/DKO
Ripk1	S6	5.06
	S25	5.76
	S166	-1.21
	T169	-1.21
	S330	-0.43

During sample preparation for the LC-MS/MS experiment, phosphopeptides were selectively enriched by Ti⁴⁺ IMAC beads. This is now better explained in Fig. 1A and in the text. This experimental setup unfortunately renders the establishment of the relative frequency of modified vs unmodified peptides impossible afterwards. We however performed a new set of experiments (New Fig. 1I-L) from which we can conclude that Ser25 phosphorylation occurs in TNFR1 complex I, and that most RIPK1 associated to complex I is phosphorylated on Ser25 (See next comment).

The relative occupancy of the S25 phosphorylation, both in RIPK1 associated with TNFR1 complex I and in cytosol, is particularly important for the conclusion that IKK-mediated phosphorylation is an important checkpoint to prevent RIPK1 activation as this would require that a substantial proportion of the RIPK1 molecules within complex I are modified on this site. This would also be a relevant consideration when using S25D knockin cells where all the RIPK1 are mutated.

- The reviewer is raising an interesting point. We performed two new experiments to address the question. In the first experiment (New Fig. 1I-J), we looked at the relative occupancy of pSer25 RIPK1 vs RIPK1 in TNFR1 complex I. To do so, we performed two subsequent immunoprecipitations (IP): first of complex I (by FLAG IP, IP1) and then of pS25 RIPK1 from complex I eluted from the beads by SDS treatment (IP2). The immunoblotting of IP2 for RIPK1 demonstrated occurrence of phosphorylation of RIPK1 on Ser25 in TNFR1 complex I, validating our LC-MS/MS results. Then, by immunoblotting complex I depleted or not of pS25 RIPK1 for RIPK1, we next demonstrated that most of the RIPK1 associated to complex I is phosphorylated on Ser25. Indeed, the RIPK1 signal was greatly reduced in complex I depleted of pSer25 RIPK1 than in total complex I (“complex I pre IP2” vs “complex I post IP2”).

In a second experiment (New Fig 1K-L), we determined the relative presence of pSer25 RIPK1 in complex I vs the cytosol. To do so, we immunoprecipitated pSer25 RIPK1 from total cell lysate previously depleted or not of complex I (by FLAG IP). The immunoblotting of the pSer25 RIPK1 IP for RIPK1 clearly demonstrated that most, if not all, pSer25 RIPK1 is associated to TNFR1 complex I. Indeed, the signal for pSer25 RIPK1 was drastically reduced in the sample depleted of complex I.

Related to this, it is somewhat surprising that mice heterozygous for the S25D mutation are almost completely rescued in the in vivo TNF models (Fig. 2H-K). As it implies that the S25D mutation dominantly represses activation of WT RIPK1. It would be of interest to know if MEFs or BMDMs isolated from these mice are protected from TNF-induced cell death (in presence of IKKi +/- zVAD) and fail to activate RIPK1. If indeed the S25D mutation is dominant, in vitro kinase assays could be done with mix of WT and S25D RIPK1 to investigate if S25D mutant will prevent kinase activity of WT RIPK1.

- The extent of protection of the *Ripk1*^{+/^{S25D} mice in the TNF shock model is indeed surprising. We performed the different experiments suggested by the reviewer to evaluate if the S25D mutation is dominant or not. First, we re-isolated BMDMs and MEFs from *Ripk1*^{+/^{, *Ripk1*^{+/^{S25D} and *Ripk1*^{S25D/S25D} littermates and compared their sensitivity to TNF-induced cell death. As shown below, heterozygous *Ripk1*^{+/^{S25D} cells were significantly protected from death induced by TNF+IKKi, but not to the same extent as the homozygous *Ripk1*^{S25D/S25D} cells (New SFig 3C-D), rejecting the hypothesis that the S25D mutation is dominant.}}}}}

SFig. 3C-D

This was further demonstrated by the fact that, in contrast to the homozygous S25D mutation, the heterozygous mutation only provided partial protection to the splenomegaly developed by the *Shpn*^{cpdm/cpdm} mice (New Fig. 4E). As shown below, we also found that the heterozygous mutation only partly prevented RIPK1 activation (monitored by auto-phosphorylation on S166) in complex I in response to IKK inhibition after TNF stimulation, while the homozygous mutation completely prevented it (New Fig 5C).

Fig. 5C

Finally, we evaluated the effect of mixing recombinant RIPK1^{WT} with RIPK1^{S25D} in an *in vitro* kinase assay. We observed that the RIPK1^{S25D} mutant did not prevent kinase activity of RIPK1^{WT}. The measured activity of the mixture was the simple addition of the kinase activities of RIPK1^{WT} and RIPK1^{S25D}. Together, these results clearly demonstrate a dosage effect rather than a dominant effect of the S25D mutation.

We therefore assume that the extent of protection of the *Ripk1*^{+/*S25D*} mice in the TNF shock model originates from the fact that a minimum amount of cell death is required to significantly reduce the viability of the mice, and that this threshold is just not obtained in the heterozygous mice. When looking at the drop in body temperature caused by TNF injection, we observe that the *Ripk1*^{+/*S25D*} mice are not as protected as the *Ripk1*^{*S25D*/*S25D*} mice (Fig. 2J, 2L).

2. IKKs directly phosphorylate RIPK1 S25.

The conclusion of IKKs directly phosphorylate RIPK1 on S25 is drawn from a previous report, where S25 was one of the hit sites from a Mass Spectrometry analysis of RIPK1 phosphorylation sites by IKKs (Fig. S4C of Ref 22 of the paper). As Mass Spectrometry experiments could have false positive hits, an *in vitro* kinase assay or its similarity should be performed for validation before concluding that IKKs directly phosphorylate RIPK1.

- The only experimental set up that permits to support direct phosphorylation of a target residue by a specific kinase consists in the identification of the phosphorylated residue by LC-MS/MS analysis from a kinase assay using recombinant proteins. In our previous paper, we performed an *in vitro* kinase assay using recombinant IKK α /IKK β and RIPK1, and analyzed the residues of RIPK1 directly phosphorylated by IKK α /IKK β by LC-MS/MS analysis. This led to the identification of Ser25 of RIPK1 as a direct substrate of IKK α and IKK β . Despite the advantage of demonstrating direct phosphorylation, an *in vitro* kinase assay remains an artificial cell free system that can indeed be prone to give false positive results. We therefore now complemented our previous results with a mass spectrometry experiment in which we looked at phosphorylation of RIPK1 in TNFR1 complex I after 5' of TNF stimulation, a time point at which we know RIPK1 is phosphorylated by IKKs. We detected $\pm 54x$ more pSer25 RIPK1 in wild-type cells than in IKK α /IKK β DKO cells. We further validated the occurrence and the IKK dependency of this phosphorylation event by performing various experiments making use of a newly generated pS25 RIPK1 antibody. Altogether, we believe that our results unequivocally demonstrate direct phosphorylation of RIPK1 on Ser25 by IKKs, and do not see the added value of repeating the *in vitro* kinase assay.

3. The function of RIPK1 S25 phosphorylation.

All experiments regarding the function of S25 phosphorylation is based on S25D phospho-mimetic mutant. It is possible that the phospho-mimetic mutant alters protein function to reduce its kinase ability in a way not related to the mimicking of phosphorylation. To substantiate the functional importance of S25 phosphorylation, cells reconstituted with the RIPK1 S25A mutant should also be tested as it would be expected that they display increased cell death compared to WT.

- The reviewer raises a valid point, but to our knowledge, the generation of a phospho-mimetic mutation remains the only way to directly and specifically address the effect of the phosphorylation of a particular residue experimentally. The generation of a phospho-null mutation is certainly an interesting complementary approach, but that does not exactly provide answers to the same question.

Our results demonstrate that IKK α / β phosphorylate RIPK1 on Ser25 in response to TNF sensing (Fig. 1). The analysis of the crystal structures of RIPK1, RIPK2 and RIPK4 predicts that the addition of a phosphate group on Ser25 of RIPK1 would alter its enzymatic activity. This prediction is confirmed by performing kinase assays making use of recombinant RIPK1 carrying the phospho-mimetic S25D mutation (Fig. 6). Importantly, this effect does not result from a general disturbance in RIPK1 structure caused by the Ser25 mutation since recombinant S25A RIPK1 shows comparable enzymatic activity as recombinant WT RIPK1 (Fig. below).

In addition, S25D RIPK1 is still normally recruited to TNFR1 complex I in cells (Fig. 5), where its role in contributing to MAPK and NF- κ B activation is unaltered (Fig. S3). We show that Ser25 phosphorylation of RIPK1 is defective in three conditions affecting proper IKK activation (IKK, TAK1 and Sharpin inactivation)(Fig. 1, 3, 4). In each of these conditions, defective pS25 RIPK1 is associated with increased RIPK1 activity in complex I (monitored by auto-phosphorylation on Ser166)(Fig. 5), and RIPK1 kinase-dependent cell death (Fig. 2, 3, 4). Restoring Ser25 phosphorylation of RIPK1 by expressing the phospho-mimetic S25D RIPK1 not only reversed the activation of RIPK1 in complex I (Fig. 5), but also protected the cells from TNF-induced cell death (Fig. 1-4). Together, these results allow us to conclude that phosphorylation of RIPK1 on Ser25 serves as a physiological brake directly repressing RIPK1 kinase activity.

In accordance with the reviewer's comment, our results had however not yet addressed whether the sole inhibition of Ser25 phosphorylation suffices to activate RIPK1 in complex I and to trigger RIPK1-kinase dependent cell death upon single TNF sensing. We therefore followed the reviewer's recommendation and reconstituted RIPK1-deficient MEFs with WT RIPK1 or with the S25A phospho-null mutant. We observed that preventing pS25 RIPK1 with the S25A mutation was not sufficient to activate RIPK1 in TNFR1 complex I nor to trigger cell death in response to single TNF sensing, while additional IKK inhibition similarly activated both readouts (New Fig. 7A-B). The fact that the S25A phospho-null mutation did not mimic IKK inhibition indicates that the repressive role of IKKs on RIPK1 is not limited to Ser25 phosphorylation, but additionally involves phosphorylation of another target, which could either be another residue of RIPK1 or another protein. It is sufficient to mimic one brake on RIPK1 (pSer25) to prevent RIPK1 activation, but all the brakes need to be released to activate RIPK1 and to induce cell death. By combining the S6A and S25A mutations, we excluded the implication of IKK-mediated phosphorylation of RIPK1 on Ser6 (New Fig. 7B). In addition, by challenging the S25A reconstituted cells with TNF in presence of CHX, we also excluded genes activated by IKK-dependent phosphorylation (New Fig. 7C).

Fig. 7A

Fig. 7B

Fig. 7C

Altogether, our results support a two-signal model, whereby IKKs inhibit RIPK1 by repressing its kinase activity through Ser25 phosphorylation and by preventing its activation through phosphorylation of a yet to be discovered additional target (New Fig. 7D-E).

Fig. 7D-E

It would also be important to demonstrate *in vitro* that S25-phosphorylated RIPK1 activity is inhibited, and its inhibition level compared to RIPK1 S25D mutant.

- The fact that the S25D mutant is protected implies that a negative charge at Ser25, either induced by phosphorylation or mimicked by an aspartate, is required to reduce RIPK1 kinase activity. We included a new Figure (New Fig. 5B) in which we provide the requested experiment performed in cells. We show that there is an inverse correlation between Ser25 phosphorylation in complex I and RIPK1 kinase activity visualized by autophosphorylation. This provides evidence that RIPK1 is kinase inactivated when it is phosphorylated on Ser25. The comparison with the S25D mutation is provided in new Fig. 5C. We would have been interested in performing *in vitro* experiments with RIPK1 specifically phosphorylated on Ser25, but we do not see any ways to generate this without affecting other phosphorylation sites on RIPK1.

Fig. 5B

Fig. 5C

In addition, the authors claim that RIPK1 S25D protects from TNF-induced cell death but the assays, in particular when assessing MEFs, are short term assays that are ended within 24 h and in some cases only 7 h (e.g. Fig 2E). Thus, it is not clear if the S25D mutation provides long-term protection from cell death or if it just delays cell death. For example, the time-dependent cell death rates shown in Figure 3B-C suggests that cell death in the S25D cells is simply delayed and would reach the same level as in WT cells if the assays was extended. Yet, the authors state that the S25D mutation "greatly protects from cell death induced by TNF when TAK1 is inhibited". It would be appropriate (and more accurate) to specify in the results section and discussion if the mutation provides long-term protection or delays cell death.

- We agree with the observation of the reviewer that it may appear like the S25D only provides short term protection in MEFs. We have noticed before that excessive amounts of TNF initially cause RIPK1-dependent cell death, but that at later time points considerable RIPK-independent cell death occurs. We therefore repeated the MEF experiments with lower and more physiological TNF concentrations (100 pg/ml instead of the original 20 ng/ml). It is obvious from these new graphs (New Fig. 2B, 2E and 3C) that the RIPK1 S25D MEFs are very well protected, even after 21-24h.

In addition, the long-term protection provided by the S25D mutation is also demonstrated in the *in vivo* models used, such as by the fact that S25D RIPK1 completely rescues the chronic TNF-driven phenotype of the *Shpn^{cpdm/cpdm}* mice.

4. S25 phosphorylation directly inhibits RIPK1 activity

All the cell death experiments are performed in mouse cells but the *in vitro* analysis of the effect of the S25D mutation on RIPK1 is performed using human RIPK1. Since the necroptosis pathway is relatively unconserved evolutionarily, the *in vitro* experiments should be performed with mouse RIPK1.

- We would like to point out that the results presented in Fig. 5 already demonstrate the effect of IKK inhibition and of mimicking Ser25 phosphorylation on the activation of mouse RIPK1 in TNFR1 complex I. As mentioned above, we have also included a new Figure (New Fig. 5B) showing inverse correlation between Ser25 phosphorylation in complex I and RIPK1 kinase activity visualized by autophosphorylation. In order to further address the reviewer's concern, we purified mouse RIPK1 to perform *in vitro* kinase assays. Unfortunately, we were not able to purify active recombinant mouse RIPK1, a problem that we had already encountered in the past. This might be a general problem since kinase assays using recombinant mouse RIPK1 have, to our knowledge, never been reported so far.

The reviewer's comment on the conservation of this regulatory mechanism through evolution is certainly a valid point. To address it, we took the reverse approach and evaluated the effect of mimicking Ser25 phosphorylation on TNF-mediated RIPK1 kinase-dependent death caused by IKK inhibition in human cells. As shown below (New Fig. 1E), human RIPK1 S25D expressing cells are also protected against TNF+IKK-induced cell death when compared to cells expressing wild-type human RIPK1.

Fig. 1E

Minor comments:

1. Recently, Xu et al. reported phosphorylation on T189 of RIPK1 by TBK1 in TNFR1-associated inflammatory complex (Xu, D. et al. (2018). TBK1 Suppresses RIPK1-Driven Apoptosis and Inflammation during Development and in Aging. *Cell*, 174(6), 1477-1491.). T189 phosphorylation site was reported to also directly inhibit RIPK1's kinase activity. It would be relevant to discuss this and comment on the relation between the two phosphorylation sites that both directly inhibit RIPK1 kinase activity.

- The two studies reporting on TBK1/IKK ϵ phosphorylation of RIPK1 (Xu et al. *Cell* 2018, Lafont et al. *Nat Cell Biol* 2018) were not published at the moment of our initial submission. The revised version of our manuscript now includes and discusses these results.

It is interesting that IKK α/β or TBK1/IKK ϵ inhibition suffices to switch the TNF response from survival to RIPK1 kinase-dependent cell death. These results indicate that these kinases cannot substitute for each other in the control of RIPK1. In line with this observation, we found that IKK α/β phosphorylate RIPK1 on Ser25 in a considerable fraction, but not in the entire pool, of RIPK1 associated to complex I. Interestingly, Lafont and colleagues identified S25 of RIPK1 as one of the sites phosphorylated by IKK ϵ . It will therefore be interesting to evaluate in the future if IKK α/β and TBK1/IKK ϵ phosphorylate identical, different or overlapping fractions of RIPK1 associated with TNFR1 complex I.

2. Line 82: The list of proteins in complex I is not complete; TAB-TAK1, the IKK complex, A20, CYLD, Spata2 also have been found in TNF pulldown experiments.

- The reviewer is right, and it even contains more proteins than those ones. Since our point is not list all the identified proteins, we have adapted the sentence to:
“RIPK1 is most extensively studied in the context of TNF signaling. Binding of TNF to TNFR1 results in the rapid assembly of a receptor-bound primary complex (complex I) that includes, among others, RIPK1, TRADD, cIAP1/2, LUBAC (composed of SHARPIN, HOIP and HOIL-1), TAB-TAK1 and the IKK complex (composed of NEMO, IKK α and IKK β).”

3. Line 127: The authors state "defects in this IKK / checkpoints presumably also explain the in vivo inflammatory phenotypes caused by..., such as in NEMO-deficient mice, SHARPIN-deficient mice or mice in which TAK1/IKKs are inhibited following Yersinia infection". It should be noted that impaired NF- κ B-mediated transcriptional response also contributes to excessive cell death, and therefore may contribute to the phenotype of the mice.

- We thank the reviewer for the remark. We have adapted the text to:
“The role of IKK α/β in repressing RIPK1 cytotoxicity is NF- κ B-independent, and its physiological importance is demonstrated by the fact that inflammatory pathologies caused by IKK α/β inactivation in mice can be driven by RIPK1 kinase-dependent cell death^{22,28}. Defects in this IKK α/β checkpoint presumably also explain, at least in part, the *in vivo* inflammatory phenotypes caused by...”

4. Line 164: The authors conclude the specificity of the protective effect of the S25D mutation only under conditions affecting IKK activation by comparing cell death percentage of WT and RIPK1 S25D cells under TNF+IKKi and TNF+CHX conditions. However, the difference between the two conditions could also be explained by the requirement of RIPK1 kinase activity only in TNF+IKKi-induced cell death (as indicated by Nec-1s inhibition).

- Thanks to the reviewer's remark, we realized that the formulation of our sentence was not very clear. We adapted it as follow:

“Importantly, the phospho-mimetic S25D mutation did not protect the cells from TNF-induced RIPK1-independent apoptosis obtained by co-stimulation with the translational inhibitor cycloheximide (CHX)^{31,32}, thereby demonstrating the specificity of the protection to conditions inducing **RIPK1 kinase-dependent cell death** (Fig. S3E-G).”

5. Line 170: Fig. 3J-K should probably be Fig. 2J-K.
6. In figure 3F, TAK1i is written as TAKi.
7. Lines 284 and 303, SHARPIN-deficient mice are referred to as Shp cpdm/cpdm, while in all other places they are referred to as Shpn cpdm/cpdm.
8. Lines 411 and 413, compound concentrations are missing for BMS-345541 and (5Z)-7-Oxozeanol.
9. Throughout the paper, certain numbers are not expressed clearly. For example, 7.5.106 (line 450) should be 7.5 106, etc. There is also a mixture of using comma and full stops as decimal points, e.g. lines 467 and 469.

➤ We thank the reviewer for noticing these mistakes. We have now corrected them.

Reviewer #2:

Overall, I find that this is an excellent and timely story which further elucidates a critical event in the regulation of TNF/RIPK1-dependent cell death. All the experiments are performed at a high technical level. I also believe this story will attract major interest. I just have a few relatively minor comments:

1. The authors should present the phospho-peptide data from Fig. 1C in a Table format. It would appear that multiple phosphorylation events were detected, but it is difficult to evaluate mass spec data using just the presented plot.

➤ Please see our answer to the main comment 1 of Reviewer #1. A table with all MS data is now provided as supplementary Table 1. All identified RIPK1 phosphosites are also now presented in New Fig. 1C.

2. S25D mutant of RIPK1 should be treated with some caution. It is clearly a kinase-inactive version of RIPK1 protein, thus its lack of function in multiple assays is expected. It certainly likely mimics a phosphorylated state of RIPK1, however, there is still some chance that the observed functional changes specifically reflect D25 residue. Thus, two critical experiments should be added with an opposite S25A mutant of RIPK1. Namely, the authors should confirm that S25A mutant shows S166 phosphorylation after stimulation with TNF alone, similar to the effect of IKKi. Further, the authors should examine whether S25A mutant renders cells sensitive to TNF alone (or perhaps to TNF in combination with MK2i).

➤ Please see our answer to the main comment 3 of Reviewer #1.

3. MK2-dependent phosphorylation of the Ser320 cluster on RIPK1 was previously reported to be critical for the survival of Yersenia-infected cells (PMID: 28920954), similar to Ser25 role. Thus, it would appear that both Ser25 and Ser320 sites ultimately function to limit the pro-death activity of RIPK1 downstream from TAK1/IKKs. It is possible that Ser25 is indeed the major site, and Ser320 cluster plays a further modulatory role. I would suggest that the authors should test a triple D mutant of S25/S321/S336 to see if this renders RIPK1 completely inactive in TNF-induced pro-death assays as opposed to some residual activity seen for S25D (for instance, in Fig. 3).

➤ The reviewer raised an interesting point. We would indeed expect that in conditions of TAK1 inhibition, in which both IKK and MK2 activity are affected, mutating both Ser25 (IKK site) and the Ser321/Ser336 (MK2 sites) to a phospho-mimetic would provide more protection to

TNF-induced cell death than the S25D phospho-mimetic alone. Unfortunately, we were unable to test this hypothesis due to the fact that the S>D/E mutations of the MK2 sites did not behave as phosphomimetics. As shown below, the S321D/E and S336D/E mutations did not give any protection to TNF+IKKi+MK2i (MK2i alone is not sufficient to switch the TNF response to death, as reported in Dondelinger et al. Nat Cell Biol 2017). The spatial charge distribution of a phospho-mimetic is different than on a phosphorylated serine, therefore not every phospho-mimetic can truly replace a phosphorylation event. We suppose this is the case for the MK2 phosphosites.

Of note, the critical role of MK2-dependent phosphorylation of RIPK1 for the survival of Yersenia-infected cells could also not be confirmed in BMDMs isolated from transgenic mice bearing a phospho-mimetic mutation in Ser321 of RIPK1 (Prof. Igor Brodsky, personal communication).

4) The authors should check whether S25D mutation has any effect on pIKK/I κ B degradation following TNF stimulation.

- This is indeed an important point. The results presented in SFig3A-B show that the S25D RIPK1 mutation has no effect on TNF-induced NF- κ B activation or MAPK signaling in MEFs or BMDMs (SFig3A-B).

Reviewer #3:

Major comments

1. The experiments were performed in mice and murine cells. The study would benefit from a comparison of the findings in human and murine cells, RIPK1 being a critical regulator of immunity in humans.

- Please see also our answer to the main comment 4 of reviewer #1. To address this question, we reconstituted RIPK1 KO human Jurkat cells with either wild-type or S25D human RIPK1. We demonstrate now that human S25D RIPK1 cells are also protected against TNF-induced cell death in comparison to their wild-type counterparts. Taking all the data together, the S25D phospho-mimetic protects against cell death in human and murine cells and impairs the kinase activity of human RIPK1 *in vitro* and the mouse RIPK1 kinase activity in cells. It therefore appears that the mechanism of regulating RIPK1 kinase activity and cell death by Ser25 phosphorylation is well conserved between human and mouse.

2. Most figures show that pSer25 does not fully prevent cell death. The pSer25-independent mechanisms of protection should be discussed more thoroughly.

- It is important to first mention that conditions preventing proper IKK activation not only affect regulation of RIPK1 by IKKs (which prevents RIPK1 kinase-dependent cell death) but also the IKK-dependent activation of the NF- κ B pathway (which prevents RIPK1-independent cell

death). It is therefore normal that mimicking pS25 does fully prevent TNF-induced cell death in IKK inactivated conditions. This being said, it is true that the extent of protection against RIPK1 kinase-dependent cell death was sometimes lower with the S25D mutation than with the K45A kinase-dead mutation or with the use of the RIPK1 inhibitor Nec-1s. This may indicate that Ser25 phosphorylation does not fully inhibit but rather greatly represses RIPK1 activity, which is in line with the residual activity observed in the kinase assay using recombinant proteins (Fig. 6D-E). Alternatively, the absence of full kinase inhibition may result from the fact that a S>D mutation does not fully reconstitute the phosphorylated state of the residue.

These two points are now discussed in the text as follow:

“The extent of protection against RIPK1 kinase-dependent cell death was sometimes lower with the S25D mutation than with the K45A kinase-dead mutation or the use of Nec-1s. This may indicate that Ser25 phosphorylation does not fully inhibit but rather greatly represses RIPK1 activity. Alternatively, the absence of full kinase inhibition may result from the fact that a S>D mutation does not fully reconstitute the phosphorylated state of the residue. Of note, the small fraction of cell death not prevented by RIPK1 kinase inhibition in our *in vitro* cell death assays originates from NF- κ B inhibition, as previously reported²²”

3. A control of IKK inhibition levels upon treatment of mice with IKK inhibitors should be provided.

- The reviewer raises a valid point. We now demonstrate inhibition of IKKs by the inhibitor in liver lysates of injected mice. IKK inhibition is demonstrated by the defect in I κ B α phosphorylation and degradation after TNF injection. These results are now presented as New Fig. 2H

4. Add a comment on the relative importance of the cell death inhibitory role of pSer25 depending on the cell type, the mouse line etc. (Figure 2B, figure 3C, figure 4B).

- The following comment has now been added:

“In general, we observed that the S25D mutation provided a better protection to the various RIPK1 kinase-dependent cell death triggers in MEFs and MDFs than in BMDMs. This may suggest additional cell-specific regulatory mechanisms. This difference may however also originate from the *in vitro* setting since the S25D mutation completely rescued the inflammatory phenotype of the *Shpn*^{cpdm/cpdm} mice, which originates from the death of cells of various types.”

5. Add a statistical analysis of the results shown in figure 6D.

- We thank the reviewer for noticing that a statistical analysis was missing. It has now been added to the *in vitro* kinase assay.

Other comments

Line 189: Gram

Line 300: change intentionally/unintentionally

Line 342: almost as potent is too strong

Line 432: 0.1%

Lines 432 and 444: the concentration of P/S is very low

Lines 434 and 439: add references for these protocols

Line 501: add the protocol identification number

Figure 6A: add sizes of the domains and the size of full length RIPK1

Figure 6E: Thiophosphate

- We thank the reviewer for noticing these mistakes. We have now corrected them.

Discussion: a model that could be Figure 7 or Supp figure S5 would be helpful

- We thank the reviewer for the suggestion. We have now added a model in new Figure 7D-E

Figure 1G: explain the surprising profiles of the I κ B and pI κ B blots (TNF 10 and 20 min)

- I κ B α is phosphorylated before degradation. At 10' and 20', all I κ B α is degraded so there is also no phosphorylated I κ B α . From 40' on, I κ B α is resynthesized.

Figure 3K: the increase in CFU of RIPK1S25D mice is quite small. It would be interesting to measure bacterial burden in other organs.

- We have repeated the experiment once again and combined the different results. We now also provide the CFU values for the liver and for the spleen. The increase in data points improved the statistical significance. These results are presented as new Fig. 3K-L

Fig. 3K

Fig.3L

Figure S1: low levels of RIPK1 are surprisingly detected in the RIPK1S25A mutant after pS25 immunoprecipitation. How was the specificity of the antibody tested?

- The anti-pSer25 mRIPK1 antibody is a rabbit polyclonal custom-made antibody by ThermoFisher Scientific that has been purified by affinity purification/negative adsorption. The specificity of the phospho-antibody was first tested at ThermoFisher Scientific, where they performed ELISAs with the Ab against both the phosphorylated and non-phosphorylated form of RIPK1. In addition, we further confirmed its specificity using RIPK1 KO MEFs reconstituted with WT RIPK1 or with the S25A mutant. As shown in New Fig. S1B, the Ab barely immunoprecipitated any RIPK1 in the S25A RIPK1 expressing cells, while it nicely immunoprecipitated pS25 in response to TNF stimulation in the WT RIPK1 expressing cells, demonstrating much greater affinity for the pSer25 epitope.

REVIEWERS' COMMENTS:

Reviewer #1 (Remarks to the Author):

The authors have addressed the main points I had raised, and have added several additional experimental figures that strengthen the main conclusions of the study. I have no further concerns and wish to congratulate the authors on this excellent study.

Reviewer #2 (Remarks to the Author):

The authors addressed the concerns that I had in the initial review. Although the lack of S25A phenotype is a little concerning, but the sum of the data presented in the paper supports the authors' "two-hit" model whereby Ser25 phosphorylation may be sufficient but not necessary for IKK-mediated blockade of cell death. Thus, I recommend the manuscript for publication.

Reviewer #3 (Remarks to the Author):

This very well performed study has been further improved by addition of an impressive amount of clear results.

REVIEWERS' COMMENTS:

Reviewer #1 (Remarks to the Author):

The authors have addressed the main points I had raised, and have added several additional experimental figures that strengthen the main conclusions of the study. I have no further concerns and wish to congratulate the authors on this excellent study.

Reviewer #2 (Remarks to the Author):

The authors addressed the concerns that I had in the initial review. Although the lack of S25A phenotype is a little concerning, but the sum of the data presented in the paper supports the authors' "two-hit" model whereby Ser25 phosphorylation may be sufficient but not necessary for IKK-mediated blockade of cell death. Thus, I recommend the manuscript for publication.

Reviewer #3 (Remarks to the Author):

This very well performed study has been further improved by addition of an impressive amount of clear results.

Response: We are very grateful to the reviewers for their constructive criticisms during the first round of revisions which definitely improved the quality of our work. We are happy to hear that our manuscript is now deemed acceptable for publication.